

# Finite temperature and quench dynamics in the Transverse Field Ising Model from form factor expansions

**Etienne Granet[1]⋆, Maurizio Fagotti[2] and Fabian H. L. Essler[1]**

**1** The Rudolf Peierls Centre for Theoretical Physics, Oxford, University, Oxford OX1 3PU, UK
**2** LPTMS, CNRS, Université Paris Sud, Université Paris-Saclay, 91405 Orsay, France

⋆ etienne.granet@physics.ox.ac.uk

## Abstract

We consider the problems of calculating the dynamical order parameter two-point function at finite temperatures and the one-point function after a quantum quench in the transverse field Ising chain. Both of these can be expressed in terms of form factor sums in the basis of physical excitations of the model. We develop a general framework for carrying out these sums based on a decomposition of form factors into partial fractions, which leads to a factorization of the multiple sums and permits them to be evaluated asymptotically. This naturally leads to systematic low density expansions. At late times these expansions can be summed to all orders by means of a determinant representation. Our method has a natural generalization to semi-local operators in interacting integrable models.



# 1 Introduction

As a consequence of the existence of extensive numbers of conservation laws with local densities the dynamical properties of quantum integrable models at finite energy densities are both rich and unusual. The two main settings of interest are finite temperature equilibrium response and time evolution after quantum quenches. In the first setting the aim is to determine two-point functions of the form

$$\chi_{\mathcal{AB}}(x,t) = \frac{1}{Z(\beta)} \mathrm{Tr}\left[ e^{-\beta H} \mathcal{A}(x,t)\mathcal{B}(0,0) \right], \tag{1}$$

where $Z(\beta) = \mathrm{Tr}(e^{-\beta H})$ and $\mathcal{A}(x,t)$ is a Heisenberg picture operator, while in the quench setting one is interested in equal time expectation values

$$\langle \Psi | \mathcal{O}(x,t) | \Psi \rangle, \tag{2}$$

where $|\Psi\rangle$ is an initial state that is a linear superposition of an exponentially (in system size) large number of energy eigenstates.

## 1.1 Finite temperature dynamics

Early work on determining (1) focussed on the spin-1/2 XY chain in a magnetic field, which can be mapped to a non-interacting model of free fermions [1]. In [2,3] it was shown that two-point functions fulfil systems of nonlinear differential equations, which in the transverse-field Ising limit can be efficiently solved numerically [4]. The dynamics at the Ising critical point was obtained in [5,6]. The long time and distance asymptotics of two-point functions in the XX limit was obtained from the solution of a Riemann-Hilbert problem in [7] arising from a Fredholm determinant representation [8,9]. A Fredholm determinant representation was also derived for the Ising field theory [10]. A semiclassical approach to the low temperature regime in interacting integrable models was pioneered in [11] and has proved very useful [12–14] due to its relative simplicity. It is however limited in that it applies only to very low temperatures and cannot be easily extended. Perhaps the most direct approach to evaluating (1) or (2) is by introducing spectral representations, e.g.

$$\chi_{\mathcal{AB}}(\ell,t) = \frac{1}{Z(\beta)} \sum_{n,m} e^{-\beta E_n} \langle n | \mathcal{A}(0,0) | m \rangle \langle m | \mathcal{B}(0,0) | n \rangle \, e^{it(E_n - E_m) - i\ell(P_n - P_m)}, \tag{3}$$

where $|n\rangle$ are normalized eigenstates of energy $E_n$ and momentum $P_n$. Early investigations of (3) focussed on integrable quantum field theories in the infinite volume [15–20], where the spectral representations need to be regularized. This problem was solved in [21–23] and a systematic low temperature expansion of dynamical two point functions in Fourier space was obtained [23–26]. For some correlators this expansion exhibits divergences close to the zero

temperature mass shell and needs to be summed to all orders is an open problem. A similar approach was formulated for the case of the Ising field theory in [27] and used to obtain the late-time asymptotic behaviour of the order parameter two-point function [28].

In order to go beyond the low temperature regime in interacting integrable models it is useful to work in the micro-canonical ensemble and employ typicality ideas. This provides a more efficient spectral representation of the form

$$\chi_{\mathcal{AB}}(\ell, t) = \sum_m \langle E_\beta | \mathcal{A}(0,0) | m \rangle \langle m | \mathcal{B}(0,0) | E_\beta \rangle \; e^{it(E_\beta - E_m) - i\ell(P_\beta - P_m)}, \tag{4}$$

where $E_\beta$ is a typical energy eigenstate at the energy density corresponding to inverse temperature $\beta$ [9, 29]. The representation (4) can be analyzed numerically for finite systems [29]. Moreover, in particular limiting cases it appears to be very efficient in that only a small number of states need to be summed over [34]. In the zero temperature case it has proved possible to formulate, and evaluate asymptotically, a form factor expansion in the thermodynamic limit [30–33]. Very recently an axiomatic approach aimed at extending these ideas to formulate form factors between states at finite energy densities in the infinite volume limit was proposed [35] and used to formulate a spectral representation. Using this representation to obtain explicit results for dynamical two-point functions remains an open problem.

An alternative approach to finite temperature dynamics is based on the Quantum Transfer Matrix approach [36, 37]. The latter is highly efficient for determining static properties [38–42] and can be extended to dynamical correlation functions [43]. Very recently this method has been successfully applied to the XX model [44–46] and state-of-the-art results have been obtained. The generalization to determine dynamical two-point functions in interacting integrable models is an open problem.

The late time asymptotics of certain finite temperature two-point functions can also be accessed by applying generalized hydrodynamics [48, 49] to the linear response regime, see [50, 51].

## 1.2 Quench dynamics

Early work on quench dynamics again focussed on models that can be analyzed by means of free fermion techniques [52–60]. Notably, in [57,58] exact results for the late time behaviour of one and two point functions of the order parameter in the transverse field Ising model (TFIM) after quantum quenches were obtained. One way of going beyond free theories is to employ a spectral representation

$$\langle \Psi | \mathcal{O}(x,t) | \Psi \rangle = \sum_{n,m} \langle \Psi | n \rangle \langle m | \Psi \rangle \langle n | \mathcal{O}(0,0) | m \rangle \; e^{i(E_n - E_m)t - i(P_n - P_m)x} \; . \tag{5}$$

This was used to obtain the late time behaviour for small quenches in the TFIM [57,58,61–64] and the sine-Gordon model [65–67]. The small quench regime is also accessible by semiclassical methods [68–72], which have the advantage of being significantly simpler to implement. A much more efficient spectral representation is provided by the Quench Action Approach [73]. For translationally invariant initial states this allows one to express expectation values of local operators after a quantum quench from an initial state $|\Psi\rangle$ as

$$\lim_{L \to \infty} \langle \mathcal{O}(t) \rangle = \lim_{L \to \infty} \left( \frac{\langle \Psi | \mathcal{O}(t) | \Phi_s \rangle}{2 \langle \Psi | \Phi_s \rangle} + \frac{\langle \Phi_s | \mathcal{O}(t) | \Psi \rangle}{2 \langle \Phi_s | \Psi \rangle} \right), \tag{6}$$

where $|\Phi_s\rangle$ is a *representative state* fixed by two requirements: first, it is a simultaneous eigenstate of the Hamiltonian and of the (quasi)local conservation laws $I^{(n)}$ of the theory under consideration, and, second, it correctly reproduces the expectation values

$$\lim_{L \to \infty} \frac{\langle \Psi | I^{(n)} | \Psi \rangle}{L} = \lim_{L \to \infty} \frac{\langle \Phi_s | I^{(n)} | \Phi_s \rangle}{L}. \tag{7}$$

Expression (7) affords a more efficient spectral representation involving only a single sum over energy eigenstates as

$$\langle \Psi | \mathcal{O}(t) | \Phi_s \rangle = \sum_n \langle \Psi | n \rangle \langle n | \mathcal{O}(0) | \Phi_s \rangle \, e^{it(E_n - E_s)} \, . \tag{8}$$

The behaviour in the steady state reached in the limit $t \to \infty$ is given by the expectation value in the representative state and this has been analyzed in a number of cases [77–82]. The time dependence is significantly more difficult to obtain. So far results are restricted to a particular one-point function for small quenches in the sine-Gordon model [65] and density correlations at late times after a quench in the repulsive Lieb-Linger model [83].

## 1.3 Local vs semi-local operators

Locality properties of the operator of interest have important implications in both finite temperature and quench contexts. For quantum quenches this was emphasized in [55, 56] and clarified through explicit calculations in Refs [57,58,65,73]. A precise definition of the mutual locality index $\omega(A, B)$ of two operators exists in the context of relativistic integrable quantum field theory, see e.g. [84]; specifically, the product of operators $A(x, \tau)B(0, 0)$ as a function of $(x, \tau)$ has the property

$$\mathcal{A}_C \big[ A(x, \tau)B(0, 0) \big] = e^{2\pi i \omega(A, B)} A(x, \tau)B(0, 0), \tag{9}$$

where $\mathcal{A}_C$ denotes the analytic continuation along a counter-clockwise contour $C$ around zero. Let us for simplicity consider the case of a diagonal scattering theory with only a single "elementary" particle excitation created by the field $\Psi(x)$. A convenient basis of energy eigenstates is given in terms of scattering states of elementary excitations

$$| \theta_1, \ldots, \theta_n \rangle \, , \tag{10}$$

where $\theta_j$ are rapidity variables related to the energy and momentum of a single-particle excitation by $\varepsilon(\theta) = M \cosh(\theta)$ , $p(\theta) = \frac{M}{v} \sinh(\theta)$. Spectral representations of correlation functions (in the infinite volume) involve form factors like

$$\langle \theta_1, \ldots, \theta_N | A(0, 0) | \theta_1', \ldots, \theta_M' \rangle \, . \tag{11}$$

As we will see below the case $M = N$ is of particular interest. *Local operators* have vanishing mutual locality index with $\Psi(x)$. As a consequence of kinematic poles [85] the form factors become singular when rapidities in the set $\{\theta_j\}$ approach those in $\{\theta_j'\}$. In the case $N = M$ the structure of singularities is [22]

$$\langle \theta_1 + \epsilon_1, \ldots, \theta_N + \epsilon_N | A(0, 0) | \theta_1, \ldots, \theta_N \rangle = \sum_{i_N = 1}^{N} \cdots \sum_{i_1 = 1}^{N} a_{i_1 \ldots i_N}(\theta_1, \ldots, \theta_N) \frac{\epsilon_{i_1} \ldots \epsilon_{i_N}}{\epsilon_1 \ldots \epsilon_N} + \ldots \tag{12}$$

In contrast, for *semi-local* operators $B$ with $\omega(B, \Psi) = 1$ one has instead

$$\langle \theta_1 + \epsilon_1, \ldots, \theta_N + \epsilon_N | A(0, 0) | \theta_1, \ldots, \theta_N \rangle \quad = \quad \frac{2^N \langle 0 | A(0, 0) | 0 \rangle}{\epsilon_1 \ldots \epsilon_N} + \ldots \tag{13}$$

This shows that form factors of such semi-local operators are much more singular than the ones for local operators. Form factors in integrable lattice models have analogous structures of singularities.

## 1.4 One and two-point functions of semi-local operators

The nature of singularities for semi-local operators (13) has been exploited previously to obtain results for 1-point functions after *small* quantum quenches [65,73]. The aim of this work is to extend this approach to general quantum quenches as well as to dynamical two-point functions at finite temperatures. We focus on the case of the order parameter in the TFIM because the form factors are particularly simple in this case. This allows us to exhibit in considerable detail which states in the respective spectral representations contribute to the late time asymptotics of one and two-point functions. These considerations can be generalized to interacting integrable models, as will be shown in a following publication.

## 1.5 Outline and summary of the main results

We conclude our introduction with an outline of the following sections and a brief summary of our key results.

- In Section 2 we briefly summarize a number of well known results on the TFIM and then define in detail the two problems we study in this paper, namely dynamical correlation functions at finite temperature and time evolution of the order parameter after a quantum quench. Although these two problems are of a very different physical nature, we explain how they can both be formulated in terms of sums over form factors and thus be addressed with similar techniques.

- In Section 3 we develop a novel framework for organizing and (analytically) carrying out the sums over form factors in both problems. It is based on a partial fraction decomposition of the form factors, which organizes the sums according to the degree of the poles the various terms exhibit, and naturally leads to an expansion of the correlation functions in terms of the density of particles $D = \int \rho(x)dx$ of the thermal/non-equilibrium stationary state of interest, where $\rho$ is its particle density. We present in detail how this calculation works at order $\mathcal{O}(D^2)$ in the case of finite temperature equilibrium dynamics. In order to make the expansion uniform in space and time it is necessary to sum certain contributions to all orders in $D$. In this way we obtain explicit expressions for the dynamical spin-spin correlation function in an arbitrary macro-state $|\phi\rangle$, in particular thermal states. For a transverse magnetic field $h < 1$ the results reads

$$\langle\phi|\sigma_\ell^x(t)\sigma_0^x(0)|\phi\rangle \approx C \exp\left(-2\int_{-\pi}^{\pi}\rho(x)(1+2\pi\rho(x))|t\varepsilon'(x)-\ell|dx\right),$$
$$C = \xi \exp\left(-2\int_{-\pi}^{\pi}\int_{-\pi}^{\pi}\frac{\rho(y)\rho'(x)}{\tan\left(\frac{x-y}{2}\right)}dxdy\right). \tag{14}$$

The various quantities $\varepsilon, \xi$ entering this expression are defined below in Section 2. This result is exact at order $\mathcal{O}(D^2)$, which means in particular that higher orders in the expansion will contribute additive terms in the exponents that involve third and higher powers of the particle density $\rho(p)$. The expansion can be pursued to higher orders in $D$ within the framework developed in Section 3.

We then turn to the time evolution of the order parameter after a quantum quench. By combining our framework for carrying out form factor sums with the quench action approach to quantum quenches [73] we obtain a systematic expansion of the order parameter one-point function in powers of the particle density.

A key insight derived from our approach is that the late time behaviour in both problems arises from processes that involve an *arbitrary* number of particle-hole excitations over

respectively the thermal and non-equilibrium steady state, but each of them is "small" in a sense that we make precise below. We argue that this is a general feature of form factor expansions involving semi-local operators, and represents a qualitative difference to the case of local operators.

- In Section 4 we return to the quantum quench problem and show how to determine the *exact* exponent that characterizes the exponential decay of the order parameter at late times. This calculation is based on approximations valid at late times that allow the spectral sum to be cast in the form of a determinant. This representation is similar to one obtained for the impenetrable Bose gas in Ref. [91]. The late time asymptotics can be extracted from the determinant representation and leads to the result

$$\langle \sigma_\ell^x \rangle = C \exp\left( \frac{|t|}{\pi} \int_0^\pi |\varepsilon'(x)| \log(1 - 4\pi\rho(x)) dx \right) + \dots ,$$ (15)

where the constant $C$ is known up to order $\mathcal{O}(D^2)$. This exponent is in agreement with the exact expression of the decay time obtained in a very different way in Ref. [58], while our result for $C$ is new.

- In Section 5 we generalize the approach of Section 4 to the case of the dynamical spin-spin correlation function in an arbitrary macro state $|\phi\rangle$ described by a density $\rho(p)$. We obtain the following expression of the late time asymptotics

$$\langle \phi | \sigma_\ell^x(t) \sigma_0^x(0) | \phi \rangle = C \exp\left( \frac{1}{2\pi} \int_{-\pi}^\pi |t\varepsilon'(x) - \ell| \log(1 - 4\pi\rho(x)) dx \right) + \dots .$$ (16)

Here the exponent represents an exact result, while the constant $C$ is again only known to order $\mathcal{O}(D^2)$. This result is to the best of our knowledge new. We compare (16) to numerically exact results obtained using the representation of the finite temperature correlator as a Pfaffian and find perfect agreement.

## 2 Transverse Field Ising Model

The TFIM Hamiltonian on a ring with $L$ sites reads

$$H(h) = -J \sum_{j=1}^L \left( \sigma_j^x \sigma_{j+1}^x + h\sigma_j^z \right) ,$$ (17)

where $\sigma_j^\alpha$ acts like the corresponding Pauli matrix at sites $j$ and like the identity elsewhere. We assume $J, h > 0$ and consider periodic boundary conditions. We refer the reader to Appendix A of [57] for details about the diagonalization of this Hamiltonian. We simply recall here that it can be expressed in terms of free fermions $\alpha_k$ as

$$H(h) = \sum_k \varepsilon(k) \left( \alpha_k^\dagger \alpha_k - \frac{1}{2} \right), \qquad \varepsilon(k) = 2J\sqrt{1 + h^2 - 2h\cos k} ,$$ (18)

and that the Hilbert space is divided into a Neveu Schwartz (NS) sector with states of the form

$$|q_1 \dots q_{2n}\rangle = \alpha_{q_1}^\dagger \dots \alpha_{q_{2n}}^\dagger |0\rangle_{NS}, \qquad q_i = \frac{2\pi}{L}\left( n_i + \frac{1}{2} \right), \qquad n_i = -\frac{L}{2}, \dots, \frac{L}{2} - 1 ,$$ (19)

and a Ramond (R) sector

$$|p_1 \dots p_{2m+1}\rangle = \alpha_{p_1}^\dagger \dots \alpha_{p_{2m+1}}^\dagger |0\rangle_R, \qquad p_i = \frac{2\pi n_i}{L} \qquad n_i = -\frac{L}{2}, \dots, \frac{L}{2} - 1 ,$$ (20)

where the energy $E(\{q_i\})$ and momentum $P(\{q_i\})$ of such states are given by

$$E(\{q_i\}) = \sum_{i=1}^{2n} \varepsilon(q_i), \qquad P(\{q_i\}) = \sum_{i=1}^{2n} q_i, \tag{21}$$

with an identical relation for the Ramond sector.

We will be interested in two problems involving the summation of form factors of the order parameter over the full Hilbert space, which are given by [87–90]

$$\begin{aligned}
{}_{\mathrm{NS}}\langle q_1,...,q_{2n}|\sigma_\ell^x|p_1...p_m\rangle_{\mathrm{R}} &= e^{-i\ell(\sum_{j=1}^{2n} q_j - \sum_{l=1}^{m} p_m)} i^{\lfloor n+m/2 \rfloor} (4J^2 h)^{(m-2n)^2/4} \sqrt{\xi\xi_L} \\
&\times \prod_{j=1}^{2n}\left(\frac{e^{\eta_{q_j}}}{L\varepsilon(q_j)}\right)^{1/2} \prod_{l=1}^{m}\left(\frac{e^{-\eta_{p_l}}}{L\varepsilon(p_l)}\right)^{1/2} \prod_{j<j'}^{2n}\frac{\sin\frac{q_j-q_{j'}}{2}}{\varepsilon_{q_jq_{j'}}} \prod_{l<l'}^{m}\frac{\sin\frac{p_l-p_{l'}}{2}}{\varepsilon_{p_lp_{l'}}} \prod_{j=1}^{2n}\prod_{l=1}^{m}\frac{\varepsilon_{q_jp_l}}{\sin\frac{q_j-p_l}{2}}.
\end{aligned} \tag{22}$$

Here $\xi = |1-h^2|^{1/4}$, $m$ is even (odd) for $h<1$ ($h>1$), and

$$\varepsilon_{ab} = \frac{\varepsilon(a)+\varepsilon(b)}{2}. \tag{23}$$

The terms $\xi_L$ and $e^{\eta_k}$ do not depend on the momenta (except $k$) and for large $L$ approach 1 with exponential accuracy

$$\xi_L \approx 1, \quad e^{\eta_k} \approx 1; \tag{24}$$

thus, they will be set to 1 in the following.

## 2.1 Quenches in the quench action framework

We consider the following quantum quench setup [57, 58]: at time $t=0$ we prepare the system in the ground state of the TFIM (17) at a magnetic field $h_0 < 1$

$$|\Psi\rangle = |0;h_0\rangle_{\mathrm{NS}}. \tag{25}$$

At times $t>0$ we evolve the system with Hamiltonian $H(h)$ with $h_0 \neq h < 1$. As $|\Psi\rangle$ is not an eigenstate of $H(h)$ this results in interesting dynamics. The order parameter one-point function at time $t>0$ is given by (6), where the representative state $|\Phi_s\rangle$ is characterized by the root density [73]

$$\begin{aligned}
\rho(k) &= \frac{1-\cos\Delta_k}{4\pi}, \\
\cos\Delta_k &= \frac{hh_0 - (h+h_0)\cos k + 1}{\sqrt{1+h^2-2h\cos k}\sqrt{1+h_0^2-2h_0\cos k}}.
\end{aligned} \tag{26}$$

For later convenience we define the density of particles in the representative state

$$\rho_Q = \int_{-\pi}^{\pi} \rho(x)dx, \tag{27}$$

where the index $Q$ stands for 'quench'. In a large finite volume $L$ we may choose [73]

$$|\Phi_s\rangle = |q_1,-q_1,...,q_N,-q_N\rangle_{\mathrm{NS}}, \tag{28}$$

where the momenta $q_i$ are distributed according to the root density $\rho(k)$. The time-evolved initial state is given by

$$|\psi(t)\rangle = \prod_{p>0}\frac{1+ie^{-2it\varepsilon(p)}K(p)\alpha_{-p}^\dagger\alpha_p^\dagger}{\sqrt{1+K^2(p)}}|0\rangle, \tag{29}$$

with $K(p) = \tan(\Delta_p/2)$.

Equation (6) thus provides the following representation for the order parameter one-point function

$$\left\langle \sigma_\ell^x(t) \right\rangle = \mathrm{Re}\left[ \sum_{M=0}^{\infty} \frac{(-1)^M}{i^{M-N}M!} \sum_{\substack{0 < p_1,\ldots,p_M \\ \in \mathbb{R}}} {}_{\mathrm{R}}\langle p_1, -p_1, \ldots, p_M, -p_M | \sigma_\ell^x | q_1, -q_1, \ldots, q_N, -q_N \rangle_{\mathrm{NS}} \right.$$
$$\left. \times \prod_{j=1}^{N} \frac{e^{-2it\varepsilon(q_j)}}{K(q_j)} \prod_{j=1}^{M} K(p_j) e^{2it\varepsilon(p_j)} \right], \tag{30}$$

that is a sum of form factors over states that are expressed in terms of pairs of momenta.

## 2.2 Dynamical correlation functions at finite temperature

In the TFIM, the density of momenta $q$ of the representative state $|E_\beta\rangle$ in (4) is

$$\rho(q) = \frac{1}{2\pi} \frac{1}{1 + e^{\beta\varepsilon(q)}} . \tag{31}$$

For later convenience we define the corresponding density

$$\rho_\beta = \int_{-\pi}^{\pi} \rho(x) dx . \tag{32}$$

In practice a representative state is constructed from $\rho(q)$ as follows. We first construct the particle counting function $z(q)$ by integrating the root density

$$z(q) = \int_{-\pi}^{q} \rho(y) dy . \tag{33}$$

We then solve the equations

$$z(q_j^{(0)}) = \frac{2\pi j}{L} , \quad j = 1, \ldots, N , \tag{34}$$

where $N$ is fixed by the requirement that $|q_j^{(0)}| \le \pi$. Finally we set

$$q_j = \frac{2\pi}{L}\left( \left\lfloor \frac{L}{2\pi} q_j^{(0)} - \frac{1}{2} \right\rfloor + \frac{1}{2} \right) , \quad j = 1, \ldots, N. \tag{35}$$

Inserting a resolution of the identity between the two spin operators in (4) leads to the following spectral representation

$$\chi^{xx}(\ell, t) = \sum_{M=0}^{+\infty} \frac{1}{M!} \sum_{\substack{p_1,\ldots,p_M \\ \in \mathbb{R}}} |\langle p_1, \ldots, p_M | \sigma_l^x | q_1, \ldots, q_N \rangle|^2 e^{it(E(\{q\})-E(\{p\}))+i\ell(P(\{p\})-P(\{q\}))} . \tag{36}$$

The terms in the sum depend on the regime of the TFIM: in the ordered phase, $h < 1$, $M$ has the same (even/odd) parity as $N$, whereas in the disordered phase, $h > 1$, it has opposite parity. Moreover, in contrast to the quench case, (36) involves modulus squares of form factors, and the intermediate states do not have a structure where momenta only appear in pairs $\{-p_i, p_i\}$.

# 3 Systematic approach to form factor expansions for semi-local operators

In this section we present a general framework for carrying out the form factor sums (30) and (36) analytically at late times $Jt \gg 1$. It is based on decomposing the form factors (22) into partial fractions so that the sums over the $p$'s decouple and can be evaluated exactly. The key observation is then that an oscillatory sum with a pole of order $d$ like $\sum_n \frac{e^{int}}{(n+1/2)^d}$ grows as $t^{d-1}$, so that the leading poles give the leading time behaviour, and the terms in the partial fraction decomposition can be organized according to the total number of poles. This naturally leads to an expansion in the number of particles per unit site – $N/L$ – in the representative state, which has already been proven very efficient for simpler quantities such as the free energy in Bethe ansatz solvable interacting models [74,75].

In Sections 3.1 and 3.2 we consider the application of this framework in the context of the finite temperature case (36), due to the more canonical sum over form factors that it involves. Since the form factors differ for $h < 1$, $h > 1$ and $h = 1$, we will treat these cases separately.

We will be interested in large time or space asymptotics of correlation functions, generically defined as requiring the phase $it(E(\{q\}) - E(\{p\})) + i\ell(P(\{p\}) - P(\{q\}))$ in (36) to be large. This is in particular the case of the large time and distance asymptotics at fixed

$$\alpha = \frac{t}{\ell} \,, \tag{37}$$

on which we will focus. However, the static correlations case $t = 0$ and large $\ell$ is also covered by our calculations; we refer the reader to Section 3.5.3 for details on this case. For later convenience we introduce the following notations

$$\overline{\varepsilon}(x) = \varepsilon(x) - \frac{x}{\alpha} \,,$$
$$v_{\max} = \max_{|x| \leq \pi} \varepsilon'(x) \,, \tag{38}$$

where $v_{\max}$ is the maximal group velocity of the elementary fermion excitations in the TFIM. According to whether there exists an $x_0$ such that $\overline{\varepsilon}'(x_0) = 0$ ('time-like region', $tv_{\max} > \ell$ for $t, \ell \geq 0$) or not ('space-like region', $tv_{\max} < \ell$ for $t, \ell \geq 0$), the next-to-leading terms in our expansions differ. We will in the following compute these terms in the space-like region. Their calculation in the time-like region is more involved and will be reported elsewhere.

In Section 3.4 we briefly present the application of our framework to the dynamics after quantum quenches.

## 3.1 Case $h < 1$: identical number of particles

In this subsection we treat the case $h < 1$, for which the sum (36) includes intermediate states with the same number $N$ of particles as the representative state. We will show in Section 3.2 that the contributions of intermediate states with different particle numbers $M \neq N$ is always subleading in time. We exploit this result right away and focus on states with $M = N$ in this subsection.

### 3.1.1 Partial fraction decomposition of form factors

We recall that the partial fraction decomposition of a ratio of two polynomials $\frac{P(X)}{\prod_i (X - x_i)^{a_i}}$ with distinct $x_i$'s is the writing

$$\frac{P(X)}{\prod_i (X - x_i)^{a_i}} = P_0(X) + \sum_i \sum_{v=1}^{a_i} \frac{B_{i,v}}{(X - x_i)^v} \,, \tag{39}$$

with $P_0(X)$ a polynomial of degree $\deg(P) - \sum_i a_i$ and $B_{i,\nu}$ independent of $X$, given by $B_{i,\nu} = \frac{1}{(a_i - \nu)!} \left( \frac{d}{dX} \right)^{a_i - \nu} (P(X)(X - x_i)^{a_i})|_{X = x_i}$.

The squared form factor appearing in (36) can be written as

$$|_{\text{NS}}\langle q_1, ..., q_N | \sigma_l^x | p_1, ..., p_N \rangle_{\text{R}}|^2 = \frac{\xi}{L^{2N}} F_{\{p_i\}}^{\{q_i\}}, \tag{40}$$

with[1]

$$F_U^V = \frac{\left| \prod\limits_{u \neq u' \in U} \sin\left( \frac{u - u'}{2} \right) \prod\limits_{v \neq v' \in V} \sin\left( \frac{v - v'}{2} \right) \right|}{\prod\limits_{u \neq v \in U, V} \sin^2\left( \frac{u - v}{2} \right)} \frac{\prod\limits_{u,v \in U,V} \varepsilon_{uv}^2}{\prod\limits_{u,u' \in U} \varepsilon_{uu'} \prod\limits_{v,v' \in V} \varepsilon_{vv'}}. \tag{41}$$

The $\varepsilon(p_1)$ factors are not polynomial in $p_1$ but are nevertheless bounded and without zeros. Seen as a function of $p_1$, the square of the form factor can thus be written $\sum_i \frac{A_i}{\sin^2\left( \frac{p_1 - q_i}{2} \right)} + \frac{B_i}{\sin\left( \frac{p_1 - q_i}{2} \right)} + C$ with $A_i, B_i$ independent of $p_1$ and $C$ a bounded function of $p_1$. Repeating the operation for the other momenta, one can write

$$|_{\text{NS}}\langle q_1, ..., q_N | \sigma_l^x | p_1, ..., p_N \rangle_{\text{R}}|^2 = \frac{\xi}{L^{2N}} \sum_{\nu_1, ..., \nu_N = 0}^2 \sum_{\{f_{\vec{\nu}}\}} \frac{\mathcal{A}(\{q\}, \{p\}, \{\nu\}, f_{\vec{\nu}})}{\prod\limits_{j=1}^N \sin^{\nu_j}\left( \frac{p_j - q_{f_{\vec{\nu}}(j)}}{2} \right)}, \tag{42}$$

where the second sum is over a complete set of functions $f_{\vec{\nu}} : \{i \in \{1, ..., N\} \,|\, \nu_i \neq 0\} \mapsto \{1, ..., N\}$, and where $\mathcal{A}(\{q\}, \{p\}, \{\nu\}, f_{\vec{\nu}})$ is a bounded function of $p_j$ if $\nu_j = 0$, and independent of $p_j$ otherwise. In Fig. 1 we show examples of such functions $f_{\vec{\nu}}$. The important feature of (42) is that each $p$ appears at most once (however, the $q$'s may appear several times).

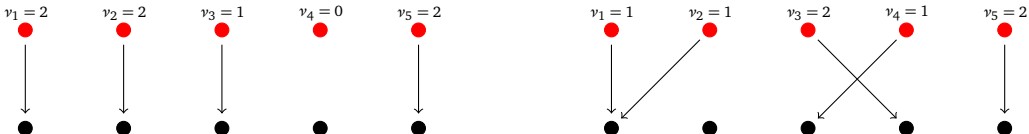

Figure 1: Sketch of two examples of a function $f$ from $p$'s in red to $q$'s in black.

### 3.1.2 Carrying the sum over the momenta $p_i$

Let us briefly anticipate the method that we will use to carry out the sum over $p$ in (36). If there is a $\nu_i = 0$ then the sum over $p_i$ is an oscillatory Riemann sum of a bounded function, hence it decays to zero with time. Thus the leading behaviour is obtained for $\nu_i > 0$ for all $i$. Then (and only in this case) the coefficients $\mathcal{A}(\{q\}, \{p\}, \{\nu\}, f_{\vec{\nu}})$ are independent of the $p$'s and the maps will no longer depend on $\{\nu_j\}$. These coefficients will be denoted by $A(\{q\}, \{\nu\}, f)$ and are obtained as

$$A(\{q\}, \{\nu\}, f) = \left[ \left( \prod_j \left( 2\frac{d}{dp_j} \right)^{2 - \nu_j} \sin^2\left( \frac{p_j - q_{f(j)}}{2} \right) \right) F_{\{p_i\}_{i=1,...,N}}^{\{q_i\}_{i=1,...,N}} \right] \Bigg|_{\{p_j = q_{f(j)}\}}. \tag{43}$$

---

[1] In $F_U^V$ the sets $U$ and $V$ can contain arbitrary momenta and are not meant to be each in the sectors $R$ and $NS$ (this freedom will be indeed useful in Section 3.1.6 below). For this reason we impose in the denominator that the momenta are different $u \neq v$, which is automatically satisfied if they are in different sectors, but not otherwise.

This follows from the partial fraction decomposition, with a factor 2 because of the 1/2 inside the sinus.

From here on we will only consider terms in the partial fraction decomposition such that $v_i > 0$ for all $i$, so that (43) applies. We denote the corresponding contribution to the spectral representation (36) of $\chi^{xx}(\ell, t)$ by $S$

$$S = \frac{\xi}{N! L^{2N}} \sum_{\substack{p_1,\dots,p_N \\ \in R}} \left\{ \left[ \sum_{v_1,\dots,v_N=1}^{2} \sum_{\{f\}} \frac{A(\{q\},\{v\},f)}{\prod_{j=1}^{N} \sin^{v_j}\left(\frac{p_j - q_{f(j)}}{2}\right)} \right] \right.$$
$$\left. \times\, e^{it(E(\{q\})-E(\{p\}))+i\ell(P(\{p\})-P(\{q\}))} \right\}, \tag{44}$$

where the third sum is over a complete set of functions $f : \{1,\dots,N\} \mapsto \{1,\dots,N\}$.

In this form one can perform the sums over $p_j$ using the following relations proven in Appendix A

$$\chi_1(q) \equiv \sum_{p \in R} \frac{e^{-it\left(\bar{\varepsilon}(p)-\bar{\varepsilon}(q)\right)}}{L \sin\left(\frac{p-q}{2}\right)} = -i\,\text{sgn}\left(t\bar{\varepsilon}'(q)\right) + \mathcal{O}(L^0 t^{-1/2})\,, \tag{45}$$

$$\chi_2(q) \equiv \sum_{p \in R} \frac{e^{-it\left(\bar{\varepsilon}(p)-\bar{\varepsilon}(q)\right)}}{L^2 \sin^2\left(\frac{p-q}{2}\right)} = 1 - \frac{2\left|t\bar{\varepsilon}'(q)\right|}{L} + \mathcal{O}(L^{-1} t^{-1/2})\,, \tag{46}$$

with $q \in NS$, to obtain

$$S = \frac{\xi}{N!} \sum_{v_1,\dots,v_N=1}^{2} \sum_{\{f\}} A(\{q\},\{v\},f) \prod_{i=1}^{N} \frac{\chi_{v_i}(q_{f(i)})}{L^{2-v_i}} e^{it(\bar{\varepsilon}(q_i)-\bar{\varepsilon}(q_{f(i)}))}\,. \tag{47}$$

Equations (46) are valid only when $\bar{\varepsilon}'(q) \neq 0$. If there is a point where $\bar{\varepsilon}'(q) = 0$, i.e. if we are in the time-like region, corrections in time to (46) have to be taken into account, and they are expected to modify significantly the subleading corrections in the correlation function. We leave this matter of discussion for future work.

### 3.1.3 Constraints on the functions $f$

The set of functions $f$ over which we need to sum is actually quite constrained. First, since $F_{\{p_i\}_{i=1,\dots N}}^{\{q_i\}_{i=1,\dots N}} = 0$ whenever $p_i = p_j$ we must have $f(i) \neq f(j)$ whenever $v_i = v_j = 2$.

We also need $f(i) \neq f(j)$ if $v_i = 1$ and $v_j = 2$. Indeed, if $f(i) = f(j)$ then in (43) there is a $\sin^2 \frac{p_i - p_j}{2}$ factor in the numerator of $F_{\{p_n\}_{n=1,\dots,N}}^{\{q_n\}_{n=1,\dots,N}}$, but as there is only one derivative with respect to $p_i$ and none with respect to $p_j$ this factor will make the coefficient $A(\{q\},\{v\},f)$ vanish upon taking $p_i = q_{f(i)} = q_{f(j)} = p_j$.

More generally, if $f$ takes $k$ times the same value at points with $v_i = 1$, then all the $k(k-1)/2$ terms $\sin^2 \frac{p_i - p_j}{2}$ contribute to a zero of order $k(k-1)$; since the number of derivatives is equal to $k$, we must have $k = 2$.

These arguments show that the sum over $f$ can be replaced by a sum over three disjoint subsets $I_0, I_1, I_2 \subset \{1,\dots,N\}$, where $I_k$ is the set of points with $v = 1$ attained $k$ times by $f$. The remaining points $\{1,\dots,N\} - (I_0 \cup I_1 \cup I_2)$ all have $v = 2$. There is a combinatorial factor

$\frac{N!}{2^{|I_2|}}$ corresponding to the number of such functions with this precise ouput. It follows that $A(\{q\}, \{v\}, f)$ depends only on the sets $I_0, I_1, I_2$ and we have

$$S = \xi \sum_{\substack{I_0, I_1, I_2 \subset \{1, \dots, N\} \\ |I_0| = |I_2|, \text{ all disjoint}}} A(I_0, I_1, I_2) 2^{-|I_2|} e^{it(\sum_{i \in I_0} \bar{\varepsilon}(q_i) - \sum_{i \in I_2} \bar{\varepsilon}(q_i))}$$

$$\times \prod_{i \in I_1} \frac{\chi_1(q_i)}{L} \prod_{i \in I_2} \frac{\chi_1^2(q_i)}{L^2} \prod_{i \notin I_{0,1,2}} \chi_2(q_i) \, . \tag{48}$$

The expression for the coefficients $A(I_0, I_1, I_2)$ can be simplified as follows. We observe that, whenever we have $v_i = 2$ in (43), the various factors depending on $p_i$ and $q_{f(i)}$ precisely compensate one another. Hence we can work with a reduced form factor involving only momenta in $I_0, I_1, I_2$

$$A(I_0, I_1, I_2) = \left[ \left( \prod_{j=1}^{n+2m} \left( 2 \frac{d}{dp_j} \right) \sin^2 \left( \frac{p_j - q_{f(j)}}{2} \right) \right) F^{\{q_i\}_{i \in I_0 \cup I_1 \cup I_2}}_{\{p_i\}_{i=1, \dots, n+2m}} \right] \Bigg|_{\{p_j = q_{f(j)}\}} , \tag{49}$$

for any function $f$ such that $\{f(i)\}_{i=1,\dots,n} = I_1$, $\{f(i)\}_{i=n+1,\dots,n+m} = I_2$,
and $\{f(i)\}_{i=n+m+1,\dots,n+2m} = I_2$. The set $I_2$ does appear twice by construction, and $I_0$ does not appear at all. The decomposition of $\{1, \dots, n + 2m\}$ into $\{1, \dots, n\}$, $\{n + 1, \dots, n + m\}$, $\{n + m + 1, \dots, n + 2m\}$ is arbitrary, and it needs only to involve one set with $n$ elements and two sets with $m$ elements.

The sum of all the terms in (48) with $|I_1| = n, |I_0| = |I_2| = m$ will be denoted by $S_{n,2m}$. We can factorize $S_{0,0}$ and, using the explicit expression for $\chi_1(q)$ and $\chi_2(q)$, write

$$S_{n,2m} = \frac{(-i)^n S_{0,0}}{(-2)^m L^{n+2m}} \sum_{\substack{I_{0,1,2} \subset \{1, \dots, N\} \\ |I_0| = |I_2| = m \\ |I_1| = n \\ \text{all disjoint}}} A(I_0, I_1, I_2) \frac{\prod_{i \in I_1} \text{sgn}(t\bar{\varepsilon}'(q_i))}{\prod_{i \in I_{0,1,2}} \left(1 - \frac{2|t\bar{\varepsilon}'(q_i)|}{L}\right)}$$

$$\times e^{it(\sum_{i \in I_0} \bar{\varepsilon}(q_i) - \sum_{i \in I_2} \bar{\varepsilon}(q_i))} \, .$$

If $n$ and $m$ stay finite in the limit $L \to \infty$ we have $\prod_{i \in I_{0,1,2}} \left(1 - \frac{2|t\bar{\varepsilon}'(q_i)|}{L}\right) = 1 + \mathcal{O}(L^{-1})$ and hence

$$S_{n,2m} = \frac{(-i)^n S_{0,0}}{(-2)^m L^{n+2m}} \sum_{\substack{q_1^0 < \dots < q_m^0 \\ q_1^1 < \dots < q_n^1 \\ q_1^2 < \dots < q_m^2 \\ \text{all distinct}}} A(\{q^0\}, \{q^1\}, \{q^2\}) e^{it \sum_{i=1}^m \bar{\varepsilon}(q_i^0) - \bar{\varepsilon}(q_i^2)} \prod_{i=1}^n \text{sgn}(t\bar{\varepsilon}'(q_i^1)) + \mathcal{O}(L^{-1}) \, . \tag{50}$$

Since the momenta selected by the sets $I_{0,1,2}$ are drawn from the momenta $\{q_j | j = 1, \dots, N\}$ of the representative state with density $\rho$, the term $S_{n,2m}$ is of order $(N/L)^{n+2m}$ times $S_{0,0}$. Hence this expansion naturally leads to an expansion in $N/L$.

### 3.1.4 Example: correlation function at $\mathcal{O}(\rho_\beta)$ uniformly in $t$ at large $t$

Let us give some examples. With $I_0 = I_1 = I_2 = \{\}$ we have $A(I_0, I_1, I_2) = 1$ hence the term

$$S_{0,0} = \xi \prod_{i=1}^N \left(1 - \frac{2\left|t\bar{\varepsilon}'(q_i)\right|}{L}\right) = \xi \exp\left(-2 \int_{-\pi}^\pi \left|t\bar{\varepsilon}'(x)\right| \rho(x) dx\right) + \mathcal{O}(L^{-1}) \, , \tag{51}$$

where, for a function $f(x)$, we used

$$\lim_{L\to\infty}\prod_{k=0}^{L-1}\left(1+\frac{f(k/L)}{L}\right)=\exp\int_0^1 f(x)dx .$$ (52)

This term is the correlation function at order 1 in $\rho$ uniformly in $t$ at large $t$.



Figure 2: Sketch of configurations contributing to $S_{0,0}$.

### 3.1.5 Example: correlation function at $\mathcal{O}(\rho_\beta^2)$

With $I_1=\{i,j\}$ and $I_0=I_2=\{\}$ we have

$$A(I_0,I_1,I_2)=\frac{2}{\sin^2\left(\frac{q_i-q_j}{2}\right)}+\frac{8\varepsilon'(q_i)\varepsilon'(q_j)}{\left(\varepsilon(q_i)+\varepsilon(q_j)\right)^2} .$$ (53)

This term leads to

$$S_{2,0}=-\frac{S_{0,0}}{L^2}\sum_{i<j}\left(\frac{2}{\sin^2\left(\frac{q_i-q_j}{2}\right)}+\frac{8\varepsilon'(q_i)\varepsilon'(q_j)}{\left(\varepsilon(q_i)+\varepsilon(q_j)\right)^2}\right)\operatorname{sgn}\left(\bar\varepsilon'(q_j)\bar\varepsilon'(q_i)\right) .$$ (54)



Figure 3: Sketch of configurations contributing to $S_{2,0}$.

For the choice $I_1=\{\}$ and $I_0=\{i\}, I_2=\{j\}$ we have

$$A(I_0,I_1,I_2)=-\frac{2}{\sin^2\left(\frac{q_i-q_j}{2}\right)} ,$$ (55)

and

$$S_{0,2}=\frac{S_{0,0}}{L^2}\sum_{i\neq j}\frac{e^{it(\bar\varepsilon(q_j)-\bar\varepsilon(q_i))}}{\sin^2\left(\frac{q_i-q_j}{2}\right)}\operatorname{sgn}\left(\bar\varepsilon'(q_j)\bar\varepsilon'(q_i)\right) .$$ (56)

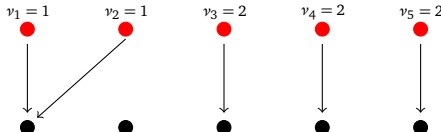

Figure 4: Sketch of configurations contributing to $S_{0,2}$.

Although they are individually both divergent in $L$ in the scaling limit $L \to \infty$, their sum is not divergent and is, see Appendix A,

$$S_{2,0} + S_{0,2} = -S_{0,0}\left(4\pi \int_{-\pi}^{\pi} \left|t\overline{\varepsilon}'(x)\right| \rho^2(x)dx + c\right), \tag{57}$$

with the following value in the space-like regime where $\operatorname{sgn}(\overline{\varepsilon}'(x))$ is constant

$$c = 2\int_{-\pi}^{\pi}\int_{-\pi}^{\pi} \frac{\rho(y)\rho'(x)}{\tan\left(\frac{x-y}{2}\right)}dxdy . \tag{58}$$

Contrarily to the previous case, this order $\rho^2$ of the correlation function at fixed large $t$ cannot be uniform in $t$, since it diverges for $t \to \infty$. In fact, the summation of other simple-pole contributions will lead to an exponentiation of this $\rho^2$ term and hence a correction of the exponent in the exponential decay.

We remark that in the very different context of the master equation approach for zero-temperature ground state correlations of a local operator in the XXZ spin chain, chains of double poles arising in some cycle integrals were observed to yield sub-leading exponential behaviours as well [76], which could suggest some yet not clear structural commonalities.

### 3.1.6 Recursive structure of $A(I_0, I_1, I_2)$

In the general case the amplitudes $A(I_0, I_1, I_2)$ are obtained from (49), but the sums over momenta associated with the index sets $I_0, I_2$ in (50) cannot be carried out as simply as in the cases treated above. In fact, as we noted earlier, the derivatives corresponding to $I_2, I_0$ in (49) have to be applied on the double zero $\sin^2\frac{p_i - p_j}{2}$ in the numerator to give a non-vanishing result, so that one actually has

$$A(I_0, I_1, I_2) = (-2)^{|I_2|}\left[\left(\prod_{i \in I_1}\left(2\frac{d}{dp_i}\right)\sin^2\frac{p_i - q_i}{2}\right)F_{\{p_i\}_{i \in I_1} \cup \{q_i\}_{i \in I_2}}^{\{q_i\}_{i \in I_1 \cup I_0}}\right]\Bigg|_{p_j = q_j, j \in I_1} . \tag{59}$$

The form factor $F_{\{p_i\}_{i \in I_1} \cup \{q_i\}_{i \in I_2}}^{\{q_i\}_{i \in I_1 \cup I_0}}$ can itself be decomposed into partial fractions. One obtains

$$A(I_0, I_1, I_2) = (-2)^{|I_2|} \sum_{\substack{\nu_i \in \{0,1,2\} \\ i \in I_1 \cup I_2 \\ \nu_i = 1 \text{ if } i \in I_1}} \sum_{\substack{f:\{i \in I_1 \cup I_2 | \nu_i > 0\} \\ \mapsto I_1 \cup I_0 \\ f(i)=i \text{ if } i \in I_1}} \frac{A(\{q_i\}_{i \in I_1 \cup I_0}, \{\nu\}, f)}{\prod_{j \in I_2} \sin^{\nu_j}\frac{q_j - q_{f(j)}}{2}} . \tag{60}$$

We observe that the sum over $\{q^2\}$ in (50) will play a similar role to the sum over $p$'s in (36), with however the important difference that they are drawn from the original $q$'s of the representative state and are not arbitrary momenta as is the case for the $p$'s in (36).

### 3.1.7 Partial fraction in $A(I_0, I_1, I_2)$ leading in density

Relation (60) reveals a recursive structure in the calculation of $A(I_0, I_1, I_2)$. However, we will not develop this recursion further here, but will rather focus on the leading partial fractions of (60) obtained with a set $\{\nu\}$ such that $\nu_i = 2$ for $i \in I_2$ and $\nu_i = 1$ for $i \in I_1$, and functions $f : I_1 \cup I_2 \mapsto I_1 \cup I_0$ that map $I_2$ to $I_0$ in a one-to-one fashion and fulfil $f(i) = i$ for $i \in I_1$.

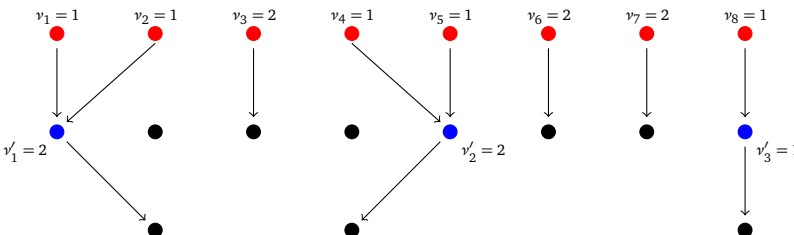

Figure 5: Sketch of the two functions $f$ after one step of recursion. In red are indicated the $p$'s, in blue the $q$'s that play the role of the $p$'s after the first step of the recursion, that are those with index in $I_1$ or $I_2$. The 'new' $q$'s on the last row are those with index in $I_1$ or $I_0$.

The coefficient then reads

$$A(\{q_i\}_{i\in I_1\cup I_0}, \{\nu\}, f) = \left[\left(\prod_{i\in I_1}\left(2\frac{d}{dp_i}\right)\sin^2\frac{p_i-q_i}{2}\right)F_{\{p_i\}_{i\in I_1}}^{\{q_i\}_{i\in I_1}}\right]\Bigg|_{p_i=q_i, i\in I_1}. \tag{61}$$

Let us select one $i \in I_1$ and introduce a reduced set $I_1' = I_1 - \{i\}$. Performing the derivative with respect to $p_i$ gives

$$A(\{q_i\}_{i\in I_1\cup I_0}, \{\nu\}, f) = \left(\prod_{k\in I_1'}\left(2\frac{d}{dp_k}\right)\sin^2\frac{p_k-q_k}{2}\right)$$
$$2\left(\sum_{k\in I_1'}\frac{1}{\tan\frac{q_i-p_k}{2}} - \frac{\varepsilon'(q_i)}{\varepsilon_{q_i p_k}} - \sum_{k\in I_1'}\frac{1}{\tan\frac{q_i-q_k}{2}} - \frac{\varepsilon'(q_i)}{\varepsilon_{q_i q_k}}\right)F_{\{p_k\}_{k\in I_1'}}^{\{q_k\}_{k\in I_1'}}\Bigg|_{p_k=q_k, k\in I_1'}. \tag{62}$$

We observe that the first factor in the second line has to be differentiated precisely one more time for the result not to vanish (the fact that $f$ is one-to-one on $I_1$ to $I_1$ is essential for this), so that

$$A(\{q_i\}_{i\in I_1\cup I_0}, \{\nu\}, f) = \sum_{j\in I_1'}\left(\frac{2}{\sin^2\frac{q_i-q_j}{2}} + 2\frac{\varepsilon'(q_i)\varepsilon'(q_j)}{\varepsilon_{q_i q_j}^2}\right)$$
$$\times\left[\left(\prod_{k\in I_1''}\left(2\frac{d}{dp_k}\right)\sin^2\frac{p_k-q_k}{2}\right)F_{\{p_k\}_{k\in I_1''}}^{\{q_k\}_{k\in I_1''}}\right]\Bigg|_{p_k=q_k, k\in I_1''}, \tag{63}$$

with $I_1'' = I_1 - \{i, j\}$. Applying the same reasoning to the remaining momenta yields

$$A(\{q_i\}_{i\in I_1\cup I_0}, \{\nu\}, f) = \sum_{P \text{ pairings of } I_1}\prod_{(i,j)\in P}\left(\frac{2}{\sin^2\frac{q_i-q_j}{2}} + 8\frac{\varepsilon'(q_i)\varepsilon'(q_j)}{(\varepsilon(q_i)+\varepsilon(q_j))^2}\right) \equiv \phi(I_1). \tag{64}$$

### 3.1.8 Result: correlation function at $\mathcal{O}(\rho_\beta^2)$ uniformly in $t$ at large $t$

In analogy with our notations for the first level of the recursive structure we denote by $S_{n,2m|0,0}$ all contributions to (50) that arise by specifying $\nu_i = 2$ for $i \in I_2$ in (60). We observe that in (59) the form factor vanishes if there are coinciding momenta in $I_0$ and in $I_2$, or momenta that

occur both in $I_1$ and $I_0$ or $I_2$. Hence one only has to impose that momenta in $I_1$ are distinct among themselves, and that momenta in $I_0$ are distinct from those of $I_2$. This gives

$$
S_{n,2m|0,0} = \frac{(-i)^n S_{0,0}}{n!(m!)^2 L^{n+2m}} \times
$$
$$
\sum_{\substack{q_1^1,\dots,q_n^1 \\ \text{all distinct}}} \sum_{\substack{q_1^0,\dots,q_m^0 \\ q_1^2,\dots,q_m^2 \\ q^0 \text{ distinct from } q^2}} \phi(\{q^1\}) \sum_{\substack{f:\{q^2\}\to\{q^0\} \\ \text{one-to-one}}} \frac{e^{it\sum_{i=1}^m \overline{\varepsilon}(q_i^0)-\overline{\varepsilon}(q_i^2)}}{\prod_{j=1}^m \sin^2 \frac{q_j^2-q_{f(j)}^0}{2}} \prod_{i=1}^n \mathrm{sgn}\,(t\overline{\varepsilon}'(q_i^1)). \tag{65}
$$

The two sets of sums factorize. We have

$$
\frac{1}{(m!)^2 L^{2m}} \sum_{\substack{q_1^0,\dots,q_m^0 \\ q_1^2,\dots,q_m^2 \\ q^0 \text{ distinct from } q^2}} \sum_{\substack{f:\{q^2\}\to\{q^0\} \\ \text{one-to-one}}} \frac{e^{it\sum_{i=1}^m \overline{\varepsilon}(q_i^0)-\overline{\varepsilon}(q_i^2)}}{\prod_{j=1}^m \sin^2 \frac{q_j^2-q_{f(j)}^0}{2}} = \frac{1}{m!}\left( \frac{1}{L^2} \sum_{q_i\neq q_j} \frac{e^{it(\overline{\varepsilon}(q_i)-\overline{\varepsilon}(q_j))}}{\sin^2 \frac{q_i-q_j}{2}} \right)^m. \tag{66}
$$

As for the terms in the sum over $\{q^1\}$, they vanish for $n$ odd, while for even $n=2p$ they are

$$
\frac{(-i)^n}{n! L^n} \sum_{\substack{q_1^1,\dots,q_n^1 \\ \text{all distinct}}} \phi(\{q^1\}) \prod_{i=1}^n \mathrm{sgn}\,(t\overline{\varepsilon}'(q_i^1))
$$
$$
= \frac{(-1)^p}{(2p)!} \frac{(2p)!}{p!2^p} \left( \frac{1}{L^2} \sum_{q_i\neq q_j} \mathrm{sgn}\,(\overline{\varepsilon}'(q_i)\overline{\varepsilon}'(q_j)) \left( \frac{2}{\sin^2 \frac{q_i-q_j}{2}} + 8\frac{\varepsilon'(q_i)\varepsilon'(q_j)}{(\varepsilon(q_i)+\varepsilon(q_j))^2} \right) \right)^p. \tag{67}
$$

It follows that the infinite volume limit of the sum of all $S_{n,2m|0,0}$ reads

$$
\sum_{n,m\geq 0} S_{n,2m|0,0} = S_{0,0} \exp\left( \frac{1}{L^2} \sum_{q_i\neq q_j} \Sigma_{ij}\, \mathrm{sgn}\,(\overline{\varepsilon}'(q_i)\overline{\varepsilon}'(q_j)) \right)
$$
$$
\Sigma_{ij} \equiv \frac{e^{it(\overline{\varepsilon}(q_i)-\overline{\varepsilon}(q_j))}-1}{\sin^2 \frac{q_i-q_j}{2}} - 4\frac{\varepsilon'(q_i)\varepsilon'(q_j)}{(\varepsilon(q_i)+\varepsilon(q_j))^2}. \tag{68}
$$

We note that it involves the same sums as in $S_{0,2}$ and $S_{2,0}$ above. We obtain

$$
\sum_{n,m\geq 0} S_{n,2m|0,0} = S_{0,0} \exp\left( -c - 4\pi \int_{-\pi}^{\pi} \rho(x)^2 |t\overline{\varepsilon}'(x)| dx + \mathcal{O}(t^{-\frac{1}{2}}) \right), \tag{69}
$$

where $S_{0,0}$ has been computed in (51). We have thus obtained the order $\mathcal{O}(\rho_\beta^2)$ contribution to the correlation function uniformly in $t$

$$
\chi^{xx}(t,\ell) \approx C \exp\left( -2\int_{-\pi}^{\pi} \rho(x)(1+2\pi\rho(x))|t\varepsilon'(x)-\ell| dx \right), \tag{70}
$$

with

$$
C = \xi \exp\left( -2\int_{-\pi}^{\pi}\int_{-\pi}^{\pi} \frac{\rho(y)\rho'(x)}{\tan\left(\frac{x-y}{2}\right)} dx dy \right). \tag{71}
$$

In the time-like region, the exponent would be the same, but the constant would differ.

This result should be compared to the semiclassical approach of Sachdev and Young [11, 86], which gives

$$\chi_{\text{SY}}^{xx}(t,\ell) \approx \xi \exp\left(-\int_{-\pi}^{\pi} \frac{dk}{\pi} e^{-\beta\varepsilon(k)} |t\varepsilon'(k) - \ell|\right). \tag{72}$$

As expected our result reduces to the semiclassical one in the limit $\beta J \gg 1$.

## 3.2 Case $h > 1$: different numbers of particles

We now turn to the case $h > 1$. Here the sum (36) involves only intermediate states with numbers of particles that are different from that of the representative state. Hence we must study form factor sums with $M \neq N$. We will compute the prefactors at order $\mathcal{O}(\rho_\beta^1)$ in the space-like region, and because of saddle points effects only at order $\mathcal{O}(\rho_\beta^0)$ in the time-like region.

### 3.2.1 General structure

We start by considering the general structure of contributions with $N \neq M$. This discussion applies also to the $h < 1$ case and in particular shows that there the dominant contributions arise from $N = M$. As in the case $M = N$, the form factor $|_{\text{NS}}\langle q_1, ..., q_N | \sigma_l^x | p_1, ..., p_M \rangle_{\text{R}}|^2$ can be decomposed into partial fractions

$$|_{\text{NS}}\langle q_1, ..., q_N | \sigma_l^x | p_1, ..., p_M \rangle_{\text{R}}|^2 = \frac{\xi (2J\sqrt{h})^{(M-N)^2}}{L^{N+M}} \sum_{\nu_1, ..., \nu_M = 0}^{2} \sum_{\{f_{\bar{\nu}}\}} \frac{A(\{q\}, \{p\}, \{\nu\}, f_{\bar{\nu}})}{\prod_{j=1}^{M} \sin^{\nu_j}\left(\frac{p_j - q_{f_{\bar{\nu}}(j)}}{2}\right)}, \tag{73}$$

with $f : \{i \in \{1, ..., M\} | \nu_i \neq 0\} \mapsto \{1, ..., N\}$ any function, and $A(\{q\}, \{p\}, \{\nu\}, f_{\bar{\nu}})$ is a bounded function of $p_j$ if $\nu_j = 0$, and independent of $p_j$ otherwise.

The important difference from the case $M = N$ is that we cannot always neglect the contributions with $\nu_j = 0$. Indeed, let us denote by $k$ the number of $\nu_j = 0$. The corresponding contributions give rise to $k$ oscillatory bounded integrals that will each decay with time. On the other hand each of the $M - k$ sums over the other momenta $p_i$ will generate an oscillating factor $e^{-it\bar{\varepsilon}(q_{f_{\bar{\nu}}(j)})}$ according to (46), while $N$ factors $e^{it\bar{\varepsilon}(q_k)}$ are already present. The resulting oscillatory sums may have singularities, but according to (46) summing these singularities does not consume any oscillatory factor, it only lowers the number of singularities. Hence in the end we will be left with $|N - M + k|$ oscillatory bounded integrals to perform. In total, there are thus $k + |N - M + k|$ of such integrals. Hence if $M \leq N$ the case $k = 0$ is still dominant at late times, but if $M > N$ then all the cases $0 \leq k \leq M - N$ are a priori of the same order. In both cases these leading terms involve $|M - N|$ oscillatory bounded integrals, so that we can conclude that the terms $M \neq N$ are exponentially smaller than the case $M = N$ in the space-like regime, and typically around[2] $t^{-|M-N|/2}$ smaller in the time-like regime.

It follows that for $h > 1$ the dominant terms in (36) are obtained for $M = N \pm 1$, which we now consider in turn.

### 3.2.2 Case $M = N - 1$

For $M = N - 1$ the dominant contribution in density in (73) is obtained with $\nu_i = 2$ for all $i = 1, ..., N - 1$. The function $f : \{1, ..., N - 1\} \mapsto \{1, ..., N\}$ has to be injective so there is one $q_i$

---

[2]Clearly, there are specific degenerate cases where the stationary phase approximation will give a different factor than $\frac{1}{\sqrt{t}}$. Due to the possible time-dependence of the integrand apart from the oscillatory term this factor may also be marginally corrected by logarithms.

not attained by $f$, leaving $(N-1)!$ possible equivalent choices for $f$ once $q_i$ is chosen. Then one has $A(\{q\},\{p\},\{v\},f_{\bar{v}}) = \frac{1}{\varepsilon(q_i)}$. Using (46), the corresponding contribution to (36) is

$$S_{0,0}^{-} = \frac{2J\xi\sqrt{h}}{L} \sum_{i=1}^{N} \frac{e^{it\bar{\varepsilon}(q_i)}}{\varepsilon(q_i)} \prod_{j\neq i} \chi_2(q_j). \tag{74}$$

This term is of order $\rho$, so in the time-like region it vanishes at the order of our computation. In the space-like region we have

$$S_{0,0}^{-} = 2J\xi\sqrt{h} \left( \int_{-\pi}^{\pi} \frac{e^{it\bar{\varepsilon}(x)}}{\varepsilon(x)} \rho(x)dx \right) \exp\left( -2\int_{-\pi}^{\pi} \left| t\bar{\varepsilon}'(x) \right| \rho(x)dx \right). \tag{75}$$

There, $\bar{\varepsilon}(x)$ is monotonous and $e^{it\bar{\varepsilon}(x)}$ is periodic (because the distance $\ell$ is an integer) so the first integral decays with time faster than any power-law and cannot be simplified further.

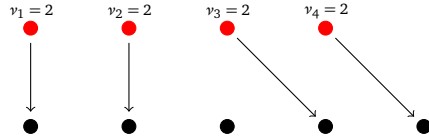

Figure 6: Sketch of the leading configurations contributing to $M = N - 1$.

### 3.2.3 Case $M = N+1$ and $k = 1$

For $M = N + 1$ with one $v_i = 0$, the dominant contribution in (73) is obtained with the remaining $v_j = 2$ for $j \neq i$. Thus $f : \{1, ..., N+1\} - \{i\} \mapsto \{1, ..., N\}$ has to be one-to-one, leaving $N!$ equivalent choices. Then one has $A(\{q\},\{p\},\{v\},f_{\bar{v}}) = \frac{1}{\varepsilon(p_i)}$. Using (46), the corresponding contribution to (36) is

$$S_{0,0}^{+} = \frac{2J\xi\sqrt{h}}{(N+1)L} \sum_{i=1}^{N+1} \sum_{p_i} \frac{e^{-it\bar{\varepsilon}(p_i)}}{\varepsilon(p_i)} \prod_{j=1}^{N} \chi_2(q_j). \tag{76}$$

In the infinite volume limit we obtain the following result in the space-like regime

$$S_{0,0}^{+} = 2J\xi\sqrt{h} \left( \int_{-\pi}^{\pi} \frac{e^{-it\bar{\varepsilon}(x)}}{2\pi\varepsilon(x)} dx \right) \exp\left( -2\int_{-\pi}^{\pi} \left| t\bar{\varepsilon}'(x) \right| \rho(x)dx \right), \tag{77}$$

where the prefactor is accurate to first order in the density $\rho_\beta$. In the time-like region the prefactor is only accurate to $\mathcal{O}(\rho_\beta^0)$, because saddle point effects arise at order $\mathcal{O}(\rho_\beta)$. A saddle point approximation gives in this regime

$$S_{0,0}^{+} = \frac{2J\xi\sqrt{h}}{\sqrt{2\pi|t|}} \left( \sum_{s\in SP} \frac{e^{\mp i\pi/4}}{\varepsilon(s)\sqrt{|\bar{\varepsilon}''(s)|}} e^{-it\bar{\varepsilon}(s)} \right) \exp\left( -2\int_{-\pi}^{\pi} \left| t\bar{\varepsilon}'(x) \right| \rho(x)dx \right), \tag{78}$$

where $SP = \{s, \bar{\varepsilon}'(s) = 0\}$ denotes the set of saddle points and $\pm$ the sign of $t\bar{\varepsilon}'(s)$.

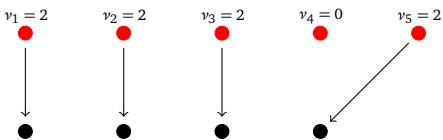

Figure 7: Sketch of the leading configurations contributing to $M = N + 1$ and $k = 1$.



### 3.2.4 Case $M = N + 1$ and $k = 0$

For $M = N + 1$ and all $v_i \neq 0$, the dominant contribution in (73) is obtained by $\{v\}, f_{\bar{v}}$ such that $v_i = v_j = 1$ with $f_{\bar{v}}(i) = f_{\bar{v}}(j)$, and the remaining $v_l = 2$ for $l \neq i, j$. Once $q_k = q_{f_{\bar{v}}(i)} = q_{f_{\bar{v}}(j)}$ are chosen, there are $\binom{N+1}{2} N!$ possibilities for $f_{\bar{v}}$. All of these lead to $A(\{q\}, \{p\}, \{v\}, f_{\bar{v}}) = \frac{-2}{\varepsilon(q_k)}$. Using (46), the corresponding contribution to (36) becomes

$$S_{0,2}^+ = -\frac{2J\xi\sqrt{h}}{L} \sum_{k=1}^{N} \frac{e^{-it\bar{\varepsilon}(q_k)}}{\varepsilon(q_k)} \chi_1^2(q_k) \prod_{j \neq k}^{N} \chi_2(q_j) . \tag{79}$$

Taking the infinite volume limit we obtain

$$S_{0,2}^+ = -2J\xi\sqrt{h} \left( \int_{-\pi}^{\pi} \frac{e^{-it\bar{\varepsilon}(x)}}{\varepsilon(x)} \chi_1^2(x)\rho(x)dx \right) \exp\left( -2 \int_{-\pi}^{\pi} \left| t\bar{\varepsilon}'(x) \right| \rho(x)dx \right) . \tag{80}$$

In the space-like region $\chi_1(x) = i \, \text{sgn}(\bar{\varepsilon}'(x))$ and one has with the prefactor at order $\rho^1$

$$S_{0,2}^+ = 2J\xi\sqrt{h} \left( \int_{-\pi}^{\pi} \frac{e^{-it\bar{\varepsilon}(x)}}{\varepsilon(x)} \rho(x)dx \right) \exp\left( -2 \int_{-\pi}^{\pi} \left| t\bar{\varepsilon}'(x) \right| \rho(x)dx \right) . \tag{81}$$

In time-like region, this term vanishes at order $\rho^0$.

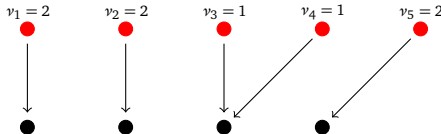

Figure 8: Sketch of the leading configurations contributing to $M = N + 1$ and $k = 0$.

### 3.2.5 Result: correlation functions for $h > 1$ at leading order in $\rho_\beta$

Putting everything together, in the space-like regime and at leading order in density we obtain the following result

$$\chi^{xx}(\ell, t) \approx 2J\xi\sqrt{h} \left[ \int_{-\pi}^{\pi} dx \frac{e^{-it\bar{\varepsilon}(x)}}{2\pi\varepsilon(x)} (1 + 4\pi\rho(x)) \right] \exp\left( -2 \int_{-\pi}^{\pi} \left| t\bar{\varepsilon}'(x) \right| \rho(x) \, dx \right), . \tag{82}$$

In the time-like regime we have instead

$$\chi^{xx}(\ell, t) \approx \frac{2J\xi\sqrt{h}}{\sqrt{2\pi|t|}} \left( \sum_{s \in SP} \frac{e^{\mp i\pi/4}}{\varepsilon(s)\sqrt{|\bar{\varepsilon}''(s)|}} e^{-it\bar{\varepsilon}(s)} \right) \exp\left( -2 \int_{-\pi}^{\pi} \left| t\bar{\varepsilon}'(x) \right| \rho(x)dx \right), \tag{83}$$

where the sum is over the saddle points $s$ of $\bar{\varepsilon}(x)$.

This result should be compared to the semiclassical approach of Sachdev and Young [11, 86], which gives

$$\chi_{SY}^{xx}(t, \ell) \approx 2J\xi\sqrt{h} \int_{-\pi}^{\pi} \frac{dx}{2\pi} \frac{e^{-it\bar{\varepsilon}(x)}}{\varepsilon(x)} \exp\left( -\int_{-\pi}^{\pi} \frac{dk}{\pi} e^{-\beta\varepsilon(k)} |t\varepsilon'(k) - \ell| \right) . \tag{84}$$

As expected our result reduces to the semiclassical one in the limit $\beta J \gg 1$.

### 3.3 Case $h = 1$

In the case $h = 1$ the structure of the form factor (22) is modified compared to the case $h \neq 1$: since $\varepsilon(0) = 0$, there are additional poles. The nature of these poles is moreover different from those appearing in the partial fraction decomposition of the previous sections, since they involve pairs of momenta: for $\varepsilon_{pp'}$ to vanish we must have $p = p' = 0$.

We note that at zero temperature the model becomes critical at $h = 1$ and correlation functions should exhibit power-law decays. These issues are beyond the scope of this work.

### 3.4 Quantum quench case

We now turn to the time evolution of the order parameter one-point function after a quench of the transverse field within the ordered phase. Our aim is to evaluate the spectral representation (30) obtained in the framework of the quench action approach.

#### 3.4.1 Generalities

The sum over form factors appearing in the context of quantum quench dynamics (30) differs from the finite temperature case (36) notably because now in both states of the form factors the momenta come in pairs $p_i, -p_i$. One can then write

$$
{}_\text{R}\langle p_1, -p_1, ..., p_N, -p_N | \sigma_\ell^x | q_1, -q_1, ..., q_N, -q_N \rangle_\text{NS} = \frac{(-4)^N \sqrt{\xi}}{L^{2N}} \prod_{j=1}^{N} \sin q_j \sin p_j
$$

$$
\times \prod_{i,j=1}^{N} \frac{\varepsilon_{q_j p_i}^4}{\varepsilon_{q_j q_i}^2 \varepsilon_{p_j p_i}^2} \frac{\prod_{i \neq j=1}^{N} (\cos q_i - \cos q_j)(\cos p_i - \cos p_j)}{\prod_{i,j=1}^{N} (\cos q_i - \cos p_j)^2} . \tag{85}
$$

Focusing again on $M = N$ in (30), we have

$$
\langle \sigma_\ell^x(t) \rangle = \text{Re} \Bigg[ \frac{\sqrt{\xi}}{N! L^{2N}} \sum_{\substack{0 < p_1, ..., p_N \\ \in \text{R}}} \prod_{j=1}^{N} 4 \frac{f(p_j)}{f(q_j)} \sin p_j \sin q_j e^{2it(\varepsilon_{p_j} - \varepsilon_{q_j})}
$$

$$
\times \prod_{i,j} \frac{\varepsilon_{q_j p_i}^4}{\varepsilon_{q_j q_i}^2 \varepsilon_{p_j p_i}^2} \frac{\prod_{i \neq j} (\cos q_i - \cos q_j)(\cos p_i - \cos p_j)}{\prod_{i,j} (\cos q_i - \cos p_j)^2} \Bigg], \tag{86}
$$

with

$$
f(p) \equiv K(p) = \sqrt{\frac{2\pi \rho(p)}{1 - (2\pi \rho(p))^2}} . \tag{87}
$$

#### 3.4.2 Differences from the finite temperature case

We apply a partial fraction decomposition to the second line of (86), seen as a ratio of polynomials in the $\cos p_j$. The procedure is the same as in Section 3.1. More precisely, we define

$$
\widetilde{F}_U^V = \frac{\left| \prod_{u \neq u' \in U} (\cos u - \cos u') \prod_{v \neq v' \in V} (\cos v - \cos v') \right|}{\prod_{u \neq v \in U, V} (\cos u - \cos v)^2} \frac{\prod_{u,v \in U, V} \varepsilon_{uv}^4}{\prod_{u,u' \in U} \varepsilon_{uu'}^2 \prod_{v,v' \in V} \varepsilon_{vv'}^2} , \tag{88}
$$

which we decompose into partial fractions with $\cos u$ taken to be the relevant variables. This gives

$$\widetilde{F}^{\{q_i\}}_{\{p_i\}} = \sum_{\nu_1,\ldots,\nu_N=0}^{2} \sum_{\{f_{\tilde{\nu}}\}} \frac{\mathcal{A}(\{q\},\{p\},\{\nu\},f_{\tilde{\nu}})}{\prod\limits_{j=1}^{N}(\cos p_j - \cos q_{f_{\tilde{\nu}}(j)})^{\nu_j}}, \tag{89}$$

where the second sum is over any function $f : \{i \in \{1,\ldots,N\}\,|\,\nu_i \neq 0\} \mapsto \{1,\ldots,N\}$, and where $\mathcal{A}(\{q\},\{p\},\{\nu\},f_{\tilde{\nu}})$ is a bounded function of $p_j$ if $\nu_j = 0$, and independent of $p_j$ otherwise.

There are however some noteworthy differences from Section 3.1 brought by the presence of factors involving the function $f(p)$ and the fact that there is still a $p_i$ dependence in the sum (86) outside the partial fraction decomposition. In place of (46) we now have

$$\frac{4}{L} \sum_{p>0,\in\mathrm{R}} \frac{\sin p \sin q' f(p)/f(q')}{\cos q - \cos p} e^{2it\varepsilon(p)} = 2i \operatorname{sgn}(t\varepsilon'(q)) \sin q' \frac{f(q)}{f(q')} e^{2it\varepsilon(q)} + \mathcal{O}(L^0 t^{-1/2})$$

$$\frac{4}{L^2} \sum_{p>0,\in\mathrm{R}} \frac{\sin p \sin q f(p)/f(q)}{(\cos q - \cos p)^2} e^{2it\varepsilon(p)} = \left(1 - \frac{4t|\varepsilon'(q)|}{L} + \frac{2i\operatorname{sgn}(\varepsilon'(q))}{L} \frac{f'(q)}{f(q)}\right) e^{2it\varepsilon(q)} \tag{90}$$

$$+ \mathcal{O}(L^{-1}t^{-1/2}),$$

as shown in Appendix A. The analog of equation (50) is given by

$$S_{n,2m} = \frac{(2i)^n S_{0,0}}{(-2)^m L^{n+2m}} \sum_{\substack{q_1^0 < \ldots < q_m^0 \\ q_1^1 < \ldots < q_n^1 \\ q_1^2 < \ldots < q_m^2 \\ \text{all distinct}}} A(\{q^0\},\{q^1\},\{q^2\}) \prod_{j=1}^{m} \left(4 \sin q_j^0 \sin q_j^2 \frac{f(q_j^2)}{f(q_j^0)}\right)$$

$$\times e^{2it \sum_{i=1}^{m} \varepsilon(q_i^2) - \varepsilon(q_i^0)} \prod_{i=1}^{n} \sin q_i^1 \operatorname{sgn}(\varepsilon'(q_i^1)), \tag{91}$$

with

$$S_{0,0} = \sqrt{\xi} \exp\left(-4 \int_0^{\pi} |t\varepsilon'(x)| \rho(x) dx\right). \tag{92}$$

Here we used that $f(p)$ (87) fulfils

$$\int_0^{\pi} \frac{f'(x)}{f(x)} \rho(x) dx = 0. \tag{93}$$

The possibility of differentiating $f(p)$ also modifies the derivation of (64), which now takes the form

$$A(\{q_i\}_{i \in I_1 \cup I_0}, \{\tilde{\nu}\}, \tilde{f}) =$$

$$\sum_{K \subset I_1} \prod_{k \in K} \frac{f'(q_k)}{f(q_k)} \sum_{P \text{ pairings of } I_1 - K} \prod_{(i,j) \in P} \left(\frac{1}{(\cos q_i - \cos q_j)^2} + 4 \frac{\partial_{\cos k}\varepsilon(q_i) \partial_{\cos k}\varepsilon(q_j)}{(\varepsilon(q_i) + \varepsilon(q_j))^2}\right). \tag{94}$$

Because of (93), however, only $K = \{\}$ remains after the sum over $q_k$.

### 3.4.3 Result: $\mathcal{O}(\rho_Q^2)$ uniformly in $t$ at large $t$

The final result of the above calculation for the time evolution of the order parameter one-point function after a quench of the transverse field within the ordered phase is

$$\langle \sigma_\ell^x(t) \rangle = C \exp\left(-4 \int_0^{\pi} \rho(x)(1 + 2\pi\rho(x))|t\varepsilon'(x)| dx\right), \tag{95}$$

where

$$C = \sqrt{\xi} \exp\left(-4 \int_0^\pi dx \int_0^\pi dy \, \rho(y)\left[\frac{\rho'(x)\sin y}{\cos y - \cos x} - 2\frac{|\varepsilon'(x)\varepsilon'(y)|}{(\varepsilon(x)+\varepsilon(y))^2}\rho(x)\right]\right). \tag{96}$$

This result holds at the second order in the density $\mathcal{O}(\rho_Q^2)$. The reader may have noted that, since $\varepsilon'(0) = \varepsilon'(\pi) = 0$, the quantum quench dynamics is a 'time-like region' case, so the saddle point effects might modify this prefactor at order $\rho^1$. It turns out, however, that $\rho(0) = \rho'(0) = \rho(\pi) = \rho'(\pi) = 0$, so the saddle point corrections are higher order in time and do not affect the prefactor.

### 3.5 Comments on the form factor summation

Having carried out form factor summations to obtain the leading late-time asymptotics in the low-density regime in both the finite temperature and the quantum quench contexts it is useful to take stock and stress some features that we expect to be of a general nature.

#### 3.5.1 Which states govern the late time dynamics?

The leading late time behaviour follows from equations (46) that enter the sum over $p$'s of the partial fraction decomposition (42). These formulas are obtained by isolating the singularity and dropping the integral of an oscillatory bounded function, *cf.* Appendix A. The singular part involves momenta $p_j$ in (36) that are at distance $\mathcal{O}(L^{-1})$ of any of the $q_i$. The aim of this section is to quantify more precisely the number of $p_j$ that have to be summed in order to recover the late time dynamics. In other words, we would like to know the smallest function $\eta(L)$ such that

$$\sum_{n=-L\eta(L)}^{L\eta(L)} \frac{e^{it(\bar{\varepsilon}(q)-\bar{\varepsilon}(p_n))}}{L^2 \sin^2\left(\frac{p_n-q}{2}\right)} = \left(1 - \frac{2\left|t\bar{\varepsilon}'(q)\right|}{L}\right) + \mathcal{O}(L^{-1}t^{-1/2}) + \mathcal{O}(L^{-2}), \tag{97}$$

with $p_n = q + \frac{2\pi}{L}(n+1/2)$. We first observe that if $L\eta(L) \to N_0$ remains finite when we take $L \to \infty$, then

$$\sum_{n=-L\eta(L)}^{L\eta(L)} \frac{e^{it(\bar{\varepsilon}(q)-\bar{\varepsilon}(p_n))}}{L^2 \sin^2\left(\frac{p_n-q}{2}\right)} \approx \sum_{n=-L\eta(L)}^{L\eta(L)} \frac{1}{L^2 \sin^2\left(\frac{p_n-q}{2}\right)} + \mathcal{O}(t^2/L^2)$$

$$\approx \sum_{n=-N_0}^{N_0} \frac{1}{\pi^2(n+1/2)^2} < \sum_{n=-\infty}^{\infty} \frac{1}{\pi^2(n+1/2)^2} = 1. \tag{98}$$

As our spectral representation involves $N \propto L$ sums of this kind we obtain an infinite product over factors that are strictly smaller than 1 and obtain a vanishing answer. Hence retaining only a finite number of $p_j$'s in our sum is clearly insufficient.

Conversely, if $\eta(L) = \eta$ stays finite we do have (97) at leading order in time because the terms we drop compared to having limits $\pm\infty$ contribute to an oscillatory integral of a bounded function

$$\int_\eta^\infty \frac{e^{itx}}{x^2}dx + \int_{-\infty}^{-\eta} \frac{e^{itx}}{x^2}dx, \tag{99}$$

that vanishes at large times.

What if now $\eta(L) \to 0$ with $L\eta(L) \to \infty$? We have, by turning the sum into an integral with Euler-Maclaurin correction terms

$$\sum_{n>\eta(L)L} \frac{e^{i\frac{\theta}{L}(n+1/2)}}{(n+1/2)^2} = \frac{1}{L} \int_\eta^\infty \frac{e^{i\theta x}}{x^2}dx + \mathcal{O}\left(\frac{1}{L^n \eta^n}, n \geq 3\right), \tag{100}$$

and

$$\int_\eta^\infty \frac{e^{i\theta x}}{x^2} dx = \frac{1}{\eta} E_2(-i\theta\eta) = \frac{i}{\eta^2\theta} e^{i\eta\theta} + \mathcal{O}\left(\frac{1}{\theta^{n+2}\eta^{n+3}}, n \ge 0\right),\tag{101}$$

obtained from the expansion of the exponential integral $E_2(x) = \int_1^\infty \frac{e^{-xu}}{u^2} du$. If we want this term to become negligible compared to (46) at late times in the scaling limit, then we need $L\eta^{n+2}(L) \to \infty$ for all $n \ge 0$ in order to ensure that both the Euler MacLaurin correction terms in (100) and (101) are negligible. Hence any power-law $\eta(L) = L^{-\nu}$ with $\nu > 0$ will not suffice. Stated differently, the number $L\eta(L)$ has to be larger than any $L^\nu$ for $0 < \nu < 1$, but any macroscopic fraction $\epsilon L$ with $\epsilon > 0$ is sufficient. We will denote by *mesoscopic* this number of states (in contrast with microscopic $\mathcal{O}(1)$ and macroscopic $\mathcal{O}(L)$).

Finally, it is clear from e.g. (51) that in order to recover an exponential decay in time, one should multiply a $\mathcal{O}(L)$ number of terms like (46), and to recover the right exponent one should take into account 'almost all' these terms at each momentum $q$.

We conclude that if the sum over the $p_j$ in (36) is viewed in terms of particle-hole excitations over the $q_i$, the leading late time behaviour emerges from a *mesoscopic* number (i.e. larger than any $L^\nu$ for $0 < \nu < 1$, but smaller than $\epsilon L$ for any $\epsilon > 0$) of particle-hole excitations *around each $q_i$*. It represents an exponential number of states, but still sub-entropic, in the sense that it includes only states whose macroscopic state is the representative state itself. We expect this to be a general feature of form factor expansions of semi-local operators that holds also in interacting theories.

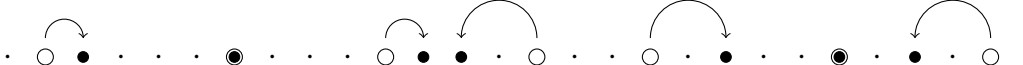

Figure 9: Sketch of the states contributing to the leading asymptotics: position of the momenta of the representative state (empty circles), position of the momenta of the intermediate state (filled circles), and position of the holes (dots).

### 3.5.2 Non-vanishing low-density limit of form factors

The leading order term both in time and density (51), obtained by keeping only the double poles in the partial fraction decomposition (42), has an interesting physical interpretation in terms of *low-density limit* of the form factor (22). This limit consists in assuming that $\rho(x)$ is small everywhere, i.e. that $L(q_{i+1} - q_i) \to \infty$ when $L \to \infty$ in the scaling limit[3]. One can observe then that the square of the form factor (40) vanishes in the scaling limit $L \to \infty$ unless each $p$ is at a distance of order $\mathcal{O}(L^{-1})$ from a $q$, and because of the low-density assumption this $q$ becomes unique in the limit $L \to \infty$. Thus one can write

$$p_i = q_i + \frac{2\pi}{L}(n_i + 1/2),\tag{102}$$

with $n_i$ an integer. In the low-density limit we have

$$\lim_{L\to\infty} \frac{\sin\frac{q_j-q_{j'}}{2}\sin\frac{p_j-p_{j'}}{2}}{\sin\frac{q_j-p_{j'}}{2}\sin\frac{q_{j'}-p_j}{2}} = -1, \qquad \lim_{L\to\infty} \frac{\varepsilon_{q_j p_{j'}}\varepsilon_{q_{j'}p_j}}{\varepsilon_{q_j q_{j'}}\varepsilon_{p_j p_{j'}}} = 1, \quad \text{for } j < j',\tag{103}$$

---

[3]Strictly speaking, this is a different limit than merely $N/L \ll 1$, since this latter one does not require $\rho(x) = \mathcal{O}(N/L)$ everywhere. For example, a ground state at zero temperature and high chemical potential would have $N/L \ll 1$ but not $\rho(0) \ll 1$.

and so in the scaling limit

$$|\langle q_1, ..., q_N | \sigma_l^x | p_1, ..., p_N \rangle|^2 = \frac{\xi}{\pi^{2N}} \prod_{j=1}^{N} \frac{1}{\left(n_j + 1/2\right)^2} \,, \tag{104}$$

which constitutes the low-density approximation of the form factor in the case it is not vanishing in the scaling limit.

Now, we trade the $1/N!$ in (36) for an ordering $p_1 < ... < p_N$ and choose an ordering $q_1 < ... < q_N$. In the low-density limit, we have $L(q_{i+1} - q_i) \to \infty$ in the large $L$ limit: Hence whenever the integers $n_i$ in (102) are $\mathcal{O}(L^0)$ the constraint on the $p's$ is automatically satisfied, and one can sum, in the large $L$ limit, on arbitrary integers. The fact that both the $1/N!$ and the constraint $p_1 < ... < p_N$ are removed in the low-density limit is a simplifying feature very specific to this regime.

Hence in the low density limit we have

$$\chi^{xx}(\ell, t) = \sum_{n_1=-\infty}^{\infty} ... \sum_{n_N=-\infty}^{\infty} \frac{\xi}{\pi^{2N}} \prod_{j=1}^{N} \frac{e^{-2i\pi t \sum_{j=1}^{N} \bar{\varepsilon}'(q_j) \frac{(n_j+1/2)}{L}}}{\left(n_j + 1/2\right)^2} \,. \tag{105}$$

This expression factorizes and can be computed along (51) with formulas (161).

### 3.5.3 Static correlations

Results (70), (82), (83) and their analogues hereafter (143), (150) for the finite temperature correlations are obtained in the regime $\ell, t \to \infty$ at fixed $\alpha = t/\ell$. In their derivation, however, we have only used that the phase $it(E(\{q\}) - E(\{p\})) + i\ell(P(\{p\}) - P(\{q\}))$ in (36) is large, therefore our calculations are also applicable to static correlations $t = 0$ at large $\ell$. It amounts to replacing $t\bar{\varepsilon}(x)$ by $-\ell x$ in all the phases $e^{-it\bar{\varepsilon}(x)}$. All the oscillatory integrals that we neglected because of their large time behaviour $\sim t^{-1/2}$ will now decay at large distances at least as $\ell^{-1}$ (and in general much faster). In particular (46) still hold, but with corrections $\mathcal{O}(L^0 \ell^{-1})$ and $\mathcal{O}(L^{-1}\ell^{-1})$ respectively. Static correlations are then obtained by replacing $|t\bar{\varepsilon}'(x)|$ (i.e., $|t\varepsilon'(x) - \ell|$) by $|\ell|$ in the final results.

## 4 Quantum quench dynamics beyond low densities

The framework presented in Section 3 is general and permits us to compute the late time behaviour and subleading corrections of form factor sums, order by order in $\rho_Q = N/L$. In particular, it yields the expression of the observables of interest as $Ce^{-t/\tau}$, where both $C$ and $\tau$ are the exact expressions at a given order (here second order) in $\rho_Q$. The computation of a generic order in the density is however rather involved. In this section and in the following one we focus on the exponent $\tau$ of the exponential decay, for which another more efficient but less general approach can be used to calculate it at all orders in $\rho_Q$, i.e. writing the correlation function as $e^{-t/\tau}$ where $\tau$ includes all orders in $\rho_Q$.

This first section treats the quantum quench problem introduced in Section 2.1.

### 4.1 Determinant representation

As shown in Section 3.2, the leading contribution in (30) is obtained for $N = M$, on which we will focus. The starting observation is that the last term of (85) can be written as a Cauchy determinant. Indeed we have

$$\frac{\prod_{i \neq j}(\cos q_i - \cos q_j)(\cos p_i - \cos p_j)}{\prod_{i,j}(\cos q_i - \cos p_j)^2} = (\det C)^2 = \det C^T C, \qquad C_{ij} = \frac{1}{\cos p_i - \cos q_j} \,. \tag{106}$$

Let us define $\bar{M} = C^T C$ and $M_{ik}^j = C_{ij} C_{kj}$, so that $\bar{M}_{ik} = \sum_j M_{ik}^j$. The determinant of $\bar{M}$ can be expanded as follows

$$
\begin{aligned}
\det \bar{M} &= \sum_{\tau \in \mathfrak{S}_N} \text{sgn}(\tau) \bar{M}_{1\tau(1)} ... \bar{M}_{N\tau(N)} \\
&= \sum_{j_1,...,j_N \in \{1,...,N\}} \sum_{\tau \in \mathfrak{S}_N} \text{sgn}(\tau) M_{1\tau(1)}^{j_1} ... M_{N\tau(N)}^{j_N} .
\end{aligned}
\tag{107}
$$

The term $M_{a\tau(a)}^{j_a} M_{b\tau(b)}^{j_b}$ is invariant under the replacement $\tau \to \tau \cdot (a\,b)$ if $j_a = j_b$, whereas the sign of the permutation changes. Hence the sum over $\tau$ vanishes unless all the $j$'s are distinct, i.e., if they are a permutation of $1, ..., N$. In conclusion we find

$$
\det \bar{M} = \sum_{\sigma \in \mathfrak{S}_N} \det M^\sigma ,
\tag{108}
$$

with $M_{ik}^\sigma = M_{ik}^{\sigma(i)}$. This relation permits one to eliminate the $1/N!$ factor in (86), which then reads

$$
\left\langle \sigma_\ell^x(t) \right\rangle = \text{Re}\left[ \frac{\sqrt{\xi}}{L^{2N}} \sum_{0 < p_1,...,p_N} \prod_{j=1}^N 4 \frac{f(p_j)}{f(q_j)} \sin p_j \sin q_j e^{2it(\varepsilon(p_j) - \varepsilon(q_j))} \prod_{i,j} \frac{\varepsilon_{q_j p_i}^4}{\varepsilon_{q_j q_i}^2 \varepsilon_{p_j p_i}^2} \det M \right],
\tag{109}
$$

where $f(p)$ is defined in (87) and $M$ is explicitly given by

$$
M_{ij} = \frac{1}{\cos q_i - \cos p_i} \frac{1}{\cos q_j - \cos p_i} .
\tag{110}
$$

### 4.1.1 Approximation

In order to proceed we now drop one of the factors in (109), which we argue is justified at late times. The factor involving the $\varepsilon_{k,k'}$ in (109) is a function

$$
g_{\{q_1,...,q_N\}}(p_1,...,p_N) = \prod_{i,j} \frac{\varepsilon_{q_j p_i}^4}{\varepsilon_{q_j q_i}^2 \varepsilon_{p_j p_i}^2} ,
\tag{111}
$$

such that for $\sigma \in \mathfrak{S}_N$

    (i) $g_{\{q_1,...,q_N\}}(p_1,...,p_N)$ is regular, symmetric and has no poles in $p_1,...,p_N$;

    (ii) $g_{\{q_1,...,q_N\}}(q_{\sigma(1)},...,q_{\sigma(N)}) = 1$;

    (iii) $\forall k = 1,...,N, \quad g_{\{q_1,...,q_N\}}(p_1,...,p_N)|_{\forall j=k+1,...,N, p_j = q_{\sigma(j)}} = g_{\{q_{\sigma(1)},...,q_{\sigma(k)}\}}(p_1,...,p_k)$;

    (iv) $\forall i = 1,...,N, \quad \partial_{p_i} g_{\{q_1,...,q_N\}}(p_1,...,p_N)|_{\forall j \, p_j = q_{\sigma(j)}} = 0$ . $\tag{112}$

The first two properties (i) and (ii) follow immediately from the definition (23). The third one (iii) means that if some $p$'s are set to some $q$'s in a one-to-one fashion, then one recovers the same function $g$ with the remaining $p$'s and $q$'s. As for property (iv), it means that (ii) holds at order $(p_i - q_{\sigma(i)})^2$.

We will now argue that by virtue of these properties setting $g$ to 1 does not affect the leading behaviour at late times $t$. First, property (i) ensures that $g$ does not modify the general structure (42) by allowing e.g. for higher order poles, or poles at other momenta. Property (ii) ensures that $g$ does not modify the values $A(I_0, I_1, I_2)$ when $I_1 = \{\}$, because in these cases there is always a double zero to differentiate and $g$ is then evaluated at a permutation of the momenta. When $I_1 \neq \{\}$, the function $g$ does change $A(I_0, I_1, I_2)$, but, because of property

(iv), it will not modify the pairing structure of (64), and will only modify the factors in (64) by an extra additive term. Finally, property (iii) allows one to repeat these steps recursively in (59). We now observe that the resulting partial fraction decomposition will always boil down to evaluating sums of the form (188). The contribution of $g$ is an additional term to (188), and so the leading time behaviour will never depend on $g$.

Based on these arguments we now make the approximation of setting $g = 1$

$$
\langle \sigma_\ell^x(t) \rangle \approx \mathrm{Re} \frac{4^N \sqrt{\xi}}{L^{2N}} \sum_{\substack{0 < p_1, \ldots, p_N \\ \in \mathrm{R}}} \det_{i,j} \left| \frac{e^{it(2\varepsilon(p_i) - \varepsilon(q_i) - \varepsilon(q_j))} \sin p_i \sin q_j f(p_i) / f(q_j)}{(\cos q_i - \cos p_i)(\cos q_j - \cos p_i)} \right|
$$
$$
= \sqrt{\xi} \det(A) ,
$$
(113)

where the matrix $A$ is given by

$$
A_{ij} = \frac{4}{L^2} \sum_{0 < p \in \mathrm{R}} \frac{e^{it(2\varepsilon(p) - \varepsilon(q_i) - \varepsilon(q_j))} \sin p \sin q_j f(p) / f(q_j)}{(\cos q_i - \cos p)(\cos q_j - \cos p)} , \quad i, j = 1, \ldots, N .
$$
(114)

We note that the above analysis is very similar to the one employed by Korepin and Slavnov in their work on the single-particle Green's function in the impenetrable Bose gas [91].

## 4.2 Asymptotic forms of the matrix elements

In the next step we work out the large-$L$ asymptotics of the matrix elements $A_{ij}$.

### 4.2.1 Diagonal matrix elements

The diagonal matrix elements were already computed in (90), *cf.* Appendix A. They read

$$
A_{ii} = 1 - \frac{|\theta_i|}{\pi} + \frac{2i \, \mathrm{sgn}(t\varepsilon'(q_i))}{L} \frac{f'(q_i)}{f(q_i)} + \mathcal{O}\left(L^{-1} t^{-1/2}\right) ,
$$
(115)

where we have defined

$$
\theta_i = \frac{4\pi t \varepsilon'(q_i)}{L} .
$$
(116)

### 4.2.2 Off-diagonal matrix elements

To compute the off-diagonal elements, we write

$$
\frac{1}{(\cos q_i - \cos p)(\cos q_j - \cos p)} = \frac{1}{\cos q_i - \cos q_j} \left( \frac{1}{\cos q_j - \cos p} - \frac{1}{\cos q_i - \cos p} \right) , \quad (117)
$$

and use the first equation in (90), see also Appendix A. We obtain

$$
A_{ij} = \frac{2i \, \mathrm{sgn}(t\varepsilon'(q_j)) \sin q_j}{L(\cos q_i - \cos q_j)} \left[ e^{it(\varepsilon(q_j) - \varepsilon(q_i))} - \frac{f(q_i) \, \mathrm{sgn}(\varepsilon'(q_j)\varepsilon'(q_i))}{f(q_j)} e^{it(\varepsilon(q_i) - \varepsilon(q_j))} \right] . \quad (118)
$$

### 4.2.3 Approximate determinant representation

Combining (115) and (118) provides the following approximate determinant representation

$$
\langle \sigma_\ell^x(t) \rangle \approx \mathrm{Re} \sqrt{\xi} \det(I - \Xi) ,
$$
(119)

where

$$\Xi_{ij} = \delta_{ij}\left[\frac{|\theta_i|}{\pi} - \frac{2i\,\mathrm{sgn}(t\varepsilon'(q_i))}{L}\frac{f'(q_i)}{f(q_i)}\right]$$
$$+ (1-\delta_{ij})\frac{2i\,\mathrm{sgn}(t\varepsilon'(q_j))\sin q_j}{L(\cos q_j - \cos q_i)}\left[e^{it(\varepsilon(q_j)-\varepsilon(q_i))} - \frac{f(q_i)\,\mathrm{sgn}(\varepsilon'(q_j)\varepsilon'(q_i))}{f(q_j)}e^{it(\varepsilon(q_i)-\varepsilon(q_j))}\right]$$
$$+ \mathcal{O}\left(L^{-1}t^{-1/2}\right). \tag{120}$$

### 4.3 Evaluating the determinant

We now write $\det M = \exp\mathrm{tr}\log M$ and use the expansion

$$\log(I-\Xi) = -\sum_{n\geq 1}\frac{\Xi^n}{n}. \tag{121}$$

#### 4.3.1 First order

The first order gives

$$\mathrm{tr}\,\Xi = 4|t|\int_0^\pi |\varepsilon'(x)|\rho(x)dx + 2i\int_0^\pi \frac{f'(x)}{f(x)}\rho(x)dx + \mathcal{O}(L^0 t^{-1/2})$$
$$= 4|t|\int_0^\pi |\varepsilon'(x)|\rho(x)dx + \mathcal{O}(L^0 t^{-1/2}). \tag{122}$$

#### 4.3.2 Second order

Since $\sum_i \Xi_{ii}^2 = \mathcal{O}\left(L^{-1}\right)$ the second order reads

$$\mathrm{tr}(\Xi^2) = \sum_{i\neq j}\Xi_{ij}\Xi_{ji} + \mathcal{O}(L^{-1}) = S_1 + S_2 + \mathcal{O}(L^{-1}), \tag{123}$$

where

$$S_1 = \frac{8}{L^2}\sum_{i\neq j}\frac{\sin q_i \sin q_j}{(\cos q_i - \cos q_j)^2}, \quad S_2 = -\frac{8}{L^2}\sum_{i\neq j}\frac{\sin q_i \sin q_j\frac{f(q_i)}{f(q_j)}}{(\cos q_i - \cos q_j)^2}e^{2it(\varepsilon(q_i)-\varepsilon(q_j))}. \tag{124}$$

These sums are computed in Appendix A. We obtain

$$\mathrm{tr}(\Xi^2) = 4|t|\int_0^\pi dx\,|\varepsilon'(x)|4\pi\rho(x)^2 + 8\int dxdy\,\frac{\sin y}{\cos y - \cos x}\rho'(x)\rho(y) + \mathcal{O}\left(t^{-1/2}\right). \tag{125}$$

Once exponentiated in the determinant, we recognize the terms obtained earlier with the partial fraction decomposition approach (95), (96), but without the contributions involving the $\varepsilon$ in the prefactor $C$. This difference is a direct consequence of our approximation $g = 1$.

#### 4.3.3 Leading late-time contribution at all orders

In order to compute the $\mathcal{O}(t)$ term at higher orders in $\rho$, we first notice that it can only arise from the second order poles in the matrix entries, hence by pairs of momenta separated by $o(L^0)$. In this regime the matrix elements become

$$\Xi_{ij} = \delta_{ij}\frac{|\theta_i|}{\pi} + \left(1-\delta_{ij}\right)\frac{-i\,\mathrm{sgn}(\theta)}{\pi}\frac{e^{i\theta_i T_i\Delta_{ij}/2} - e^{-i\theta_i T_i\Delta_{ij}/2}}{T_i\Delta_{ij}}, \tag{126}$$

with $\Delta_{ij} = i - j$. We obtain in this regime

$$
\begin{aligned}
\text{tr}(\Xi^n) &= \sum_{i_1, i_2, \dots, i_n} \Xi_{i_1 i_2} \dots \Xi_{i_n i_1} \\
&= -\sum_i \frac{1}{(i \operatorname{sgn}(\theta_i) \pi)^n T_i^n} \sum_{\Delta_1, \dots, \Delta_{n-1} \neq 0} \Big[ \prod_{m=1}^{n-1} \frac{1 - e^{-i \theta_i T_i \Delta_m}}{\Delta_m} \Big] \frac{1 - e^{i \theta_i T_i (\Delta_1 + \dots + \Delta_{n-1})}}{\Delta_1 + \dots + \Delta_{n-1}} \, .
\end{aligned}
\tag{127}
$$

The sums over $\Delta_j$ can be carried out using that for $\Delta' \neq 0$

$$
\sum_{\Delta \neq 0, -\Delta'} \frac{1 - e^{-i\theta\Delta}}{\Delta} \frac{1 - e^{i\theta(\Delta + \Delta')}}{\Delta + \Delta'} = 2i(\pi - |\theta|) \operatorname{sgn}(\theta) \frac{1 - e^{i\theta\Delta'}}{\Delta'} \, ,
\tag{128}
$$

which is obtained from $\frac{1}{\Delta(\Delta + \Delta')} = \frac{1}{\Delta'}(\frac{1}{\Delta} - \frac{1}{\Delta + \Delta'})$ and relations (161) by carefully treating the cases $\Delta = 0, -\Delta'$. Using that $\theta = \mathcal{O}(L^{-1})$ to neglect the $|\theta|$ term in (128) we arrive at

$$
\begin{aligned}
\text{tr}(\Xi^n) &= -\sum_i \frac{(2i \operatorname{sgn}(\theta_i) \pi)^{n-2}}{(i \operatorname{sgn}(\theta_i) \pi)^n T_i^n} \sum_{\Delta_1 \neq 0} \frac{(1 - e^{-i\theta_i T_i \Delta_1})(1 - e^{i\theta_i T_i \Delta_1})}{\Delta_1^2} + \mathcal{O}(L^{-1}) \\
&= \sum_i \frac{2^{n-1}}{\pi^2 T_i^n} (2 \operatorname{Li}_2(1) - \operatorname{Li}_2(e^{i\theta_i T_i}) - \operatorname{Li}_2(e^{-i\theta_i T_i})) + \mathcal{O}(L^{-1}) \\
&= \sum_i \frac{2^{n-1} |\theta_i|}{\pi T_i^{n-1}} + \mathcal{O}(L^{-1}) \\
&= 4|t| \int_0^\pi |\varepsilon'(x)| (4\pi\rho(x))^{n-1} \rho(x) dx + \mathcal{O}(L^{-1}) \, .
\end{aligned}
\tag{129}
$$

### 4.3.4 Influence of the boundaries

In the discussion above the momenta $p$ are constrained to be positive. In order to use equations (46) we therefore had to neglect possible boundary effects for $q$'s close to zero. We now verify that this does not influence the result. We have

$$
\sum_{n=-\eta_1 L}^{\eta_2 L} \frac{e^{i(n+1/2) \frac{w}{L} t}}{(n+1/2)^2} = \sum_{n=-\infty}^{\infty} \frac{e^{i(n+1/2) \frac{w}{L} t}}{(n+1/2)^2} - \frac{1}{\eta_2 L} E_2(-iw\eta_2 t) - \frac{1}{\eta_1 L} E_2(iw\eta_1 t) \, ,
\tag{130}
$$

with $E_2(x) = \int_1^\infty \frac{e^{-xt}}{t^2} dt$ the exponential integral function. $L\eta_{1,2}$ are the number of vacancies between $0, \pi$ and $q_i$. They are $\eta_1 = \frac{q_i}{2\pi}$ and $\eta_2 = \frac{\pi - q_i}{2\pi}$. Hence the correction to $\text{tr}\,\Xi$ is

$$
-\frac{2}{\pi} \int_0^\pi \left( \frac{E_2(-2i\varepsilon'(x) x t)}{x} + \frac{E_2(2i\varepsilon'(x)(\pi - x) t)}{\pi - x} \right) \rho(x) dx \, ,
\tag{131}
$$

which goes to zero for large $t$ because the density vanishes quadratically at 0 and $\pi$.

## 4.4 Result: late-time asymptotics of the order parameter after a quench

Substituting (122), (125) and (129) into (121) we arrive at the following result for the late-time asymptotics of the order parameter one-point function after a quench within the ferromagnetic phase

$$
\frac{\langle \Psi_N(t) | \sigma_\ell^x | \Psi_s(t) \rangle}{\langle \Psi_N(t) | \Psi_s(t) \rangle} = C \exp\left( \frac{|t|}{\pi} \int_0^\pi |\varepsilon'(x)| \log(1 - 4\pi\rho(x)) dx \right) \equiv C e^{-t/\tau} \, ,
\tag{132}
$$

with $C$ given in (96) at order $\mathcal{O}(\rho_Q^2)$. The decay rate reproduces the exact result obtained in [57, 58]. However, the prefactor $C$ differs from the one conjectured in [57, 58]. We address this difference in Section 4.5 below.

### 4.5 Numerical Checks

We now present some numerical checks of equation (132) for the time evolution of the order parameter. In the limit of large separations $\ell \gg 2Jt$, using the Lieb-Robinson bound and the clustering properties of the initial state $|\Psi\rangle$, the two-point function factorizes into the product of two one-point functions that are identical by translational invariance

$$\left\langle \Psi | \sigma_{\ell+1}^x(t) \sigma_1^x(t) | \Psi \right\rangle = \left\langle \Psi | \sigma_1^x(t) | \Psi \right\rangle^2 + \mathcal{O}(e^{-\gamma(\ell-2Jt)}), \tag{133}$$

with $\gamma$ a constant of order 1. We can then obtain the one-point function $\left\langle \Psi | \sigma_1^x(t) | \Psi \right\rangle$ as the square root of the two-point function $\left\langle \sigma_{\ell+1}^x(t) \sigma_1^x(t) \right\rangle$ in the limit $\ell \gg 2Jt$, which can be efficiently computed numerically, as it can be expressed as the determinant of a block Toeplitz matrix even in the thermodynamic limit [58].

Numerical checks of the exponential decay in (132) have already been reported in [58]. Since our prediction (96) differs from the one conjectured in [58], we will focus on the prefactor $\mathcal{C}_{\text{FF}}^x(\alpha)$ of the asymptotic behaviour of the two-point function in the limit $\ell, t \to \infty$, $\alpha = t/\ell$ fixed

$$\left\langle \Psi | \sigma_{\ell+1}^x(t) \sigma_1^x(t) | \Psi \right\rangle \simeq \mathcal{C}_{\text{FF}}^x(\alpha) \exp\left[\ell \int_0^\pi \frac{dk}{\pi} \log|\cos \Delta_k| \, \theta_H\left(2\varepsilon_h'(k)t - \ell\right)\right]$$

$$\times \exp\left[2t \int_0^\pi \frac{dk}{\pi} \varepsilon_h'(k) \log|\cos \Delta_k| \, \theta_H\left(\ell - 2\varepsilon_h'(k)t\right)\right]. \tag{134}$$

In [58] it was assumed that the constant $C_{\text{FF}}^x(\alpha)$ is independent of $\alpha$. Calculating the asymptotics of the correlator for $\alpha \to \infty$ then leads to [59]

$$C_{\text{FF}}^x(\infty) = \frac{1 - hh_0 + \sqrt{(1-h^2)(1-h_0^2)}}{2\sqrt{1-hh_0}\sqrt[4]{1-h_0^2}} \,. \tag{135}$$

From (96) it however follows $C \neq \sqrt{C_{\text{FF}}^x(\infty)}$, suggesting in turn that $C_{\text{FF}}^x(\alpha)$ is in fact $\alpha$-dependent. This is indeed supported by our numerical results, even though the difference $|C - \sqrt{C_{\text{FF}}^x(\infty)}|$ is tiny. In Figs 10 we show that our results (132) and (96) are in agreement with numerical calculations of the order parameter one-point function.

## 5 Dynamical correlation functions at arbitrary finite temperatures

In this section we follow the same reasoning as in Section 4 but for dynamical correlation functions at finite temperature. We again treat the two cases $h < 1$ and $h > 1$ separately.

### 5.1 Ordered phase $h < 1$

In the ordered phase, the sum in (36) involves states with the same number of particles as in the quench case (30). However, it is not possible to express each term as a Cauchy determinant as in Section 4. This can nevertheless be overcome for the leading late time behaviour, where one can work with an approximate version of the form factor (22).

We focus again on intermediate states with $M = N$ in (36), which were argued in Section 3.2 to give the leading late time behaviour. As discussed in Section 4.1.1, the dominant contribution to the correlation function arises from the sums (170), and the term proportional to $t$ on the right-hand side of (170) has its origin in the isolated singularity $\frac{1}{x^2}$ in $\frac{1}{\sin^2 x}$. Hence,

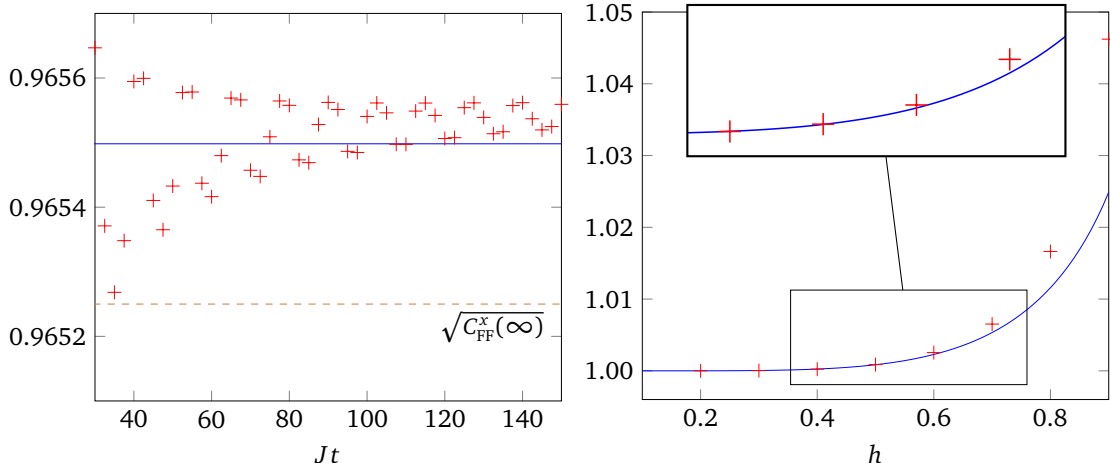

Figure 10: Left: Numerical results for $\langle \sigma_1^x(t) \rangle e^{t/\tau}$ with $\tau$ defined in (132) as a function of $t$ for a quench from $h_0 = 0.1$ to $h = 0.5$, corresponding to a density $\rho_Q = 0.0218$. The prefactor at order $\mathcal{O}(\rho_Q^2)$ (96) is shown in blue and is seen to be compatible with our numerical results. For comparison we also show $\sqrt{C_{FF}^x(\infty)} = 0.96525$. Right: Numerically determined (red) and calculated leading order (blue) prefactor (96) divided by $\sqrt{\xi}$, for quenches from $h_0 = 0.1$ to $h$ as a function of $h$. Because of the oscillations, the measured value is an average of six points between times $t = 75$ and $100$.

replacing terms of the form $\frac{1}{\sin^2 x}$ by $\frac{1}{x^2}$ in the form factor (22) will not affect the leading behaviour at late times. It follows that if we define

$$\tilde{\chi}^{xx}(\ell, t) = \frac{1}{N!} \sum_{\substack{p_1, \ldots, p_N \\ \in R}} \tilde{F}_{\{p_1, \ldots, p_N\}}^{\{q_1, \ldots, q_N\}} e^{it(E(\{q\}) - E(\{p\})) + i\ell(P(\{p\}) - P(\{q\}))} , \qquad (136)$$

$$\tilde{F}_{\{p_1, \ldots, p_N\}}^{\{q_1, \ldots, q_N\}} = \frac{\xi}{L^{2N}} \frac{\prod_{i \neq j} \frac{q_i - q_j}{2} \frac{p_i - p_j}{2}}{\prod_{i,j} \left( \frac{q_i - p_j}{2} \right)^2} , \qquad (137)$$

then we have

$$\chi^{xx}(\ell, t) = \tilde{\chi}^{xx}(\ell, t) \kappa(\ell, t) , \qquad (138)$$

with $\kappa(\ell, t)$ a function that is subleading in $\ell, t$ with respect to $\chi^{xx}(\ell, t)$.

The form factor $\tilde{F}$ has the same Cauchy determinant structure we encountered in the quench case (106). Hence we can follow through the same steps and obtain

$$\tilde{\chi}^{xx}(\ell, t) = \xi \det_{i,j=1,\ldots,N} \left[ \frac{4}{L^2} \sum_{p \in R} \frac{e^{i \frac{t}{2}(\bar{\varepsilon}(q_i) + \bar{\varepsilon}(q_j) - 2\bar{\varepsilon}(p))}}{(q_j - p)(q_i - p)} \right] . \qquad (139)$$

The leading time behaviour of the coefficients of this matrix can be computed as in Section 4.2 with formulas (161). One obtains

$$\tilde{\chi}^{xx}(\ell, t) = \xi \det(1 - \Xi) , \qquad (140)$$

with

$$
\begin{aligned}
\Xi_{ij} = {} & \delta_{ij} \frac{|2t\bar{\varepsilon}'(q_i)|}{L} \\
& + (1-\delta_{ij}) \frac{2i\,\mathrm{sgn}(t\bar{\varepsilon}'(q_j))}{L(q_j-q_i)} \left[ e^{i\frac{t}{2}(\bar{\varepsilon}(q_i)-\bar{\varepsilon}(q_j))} - \mathrm{sgn}(\bar{\varepsilon}'(q_j)\bar{\varepsilon}'(q_i)) e^{i\frac{t}{2}(\bar{\varepsilon}(q_j)-\bar{\varepsilon}(q_i))} \right] \\
& + \mathcal{O}\left(L^{-1}t^{-1/2}\right).
\end{aligned}
\tag{141}
$$

Since one is interested only in the late time dynamics, we can focus on terms $i,j$ such that $q_i - q_j = o(L^0)$ as in (126). Then one obtains the same formula as in (126)

$$
\Xi_{ij} = \delta_{ij}\frac{|\theta_i|}{\pi} + \left(1-\delta_{ij}\right)\frac{-i\,\mathrm{sgn}(\theta)}{\pi}\frac{e^{i\theta_i T_i \Delta_{ij}/2} - e^{-i\theta_i T_i \Delta_{ij}/2}}{T_i \Delta_{ij}},
\tag{142}
$$

with $\Delta_{ij} = i-j$, $T_i = \frac{1}{2\pi\rho(q_i)}$ and $\theta_i = \frac{2\pi t\bar{\varepsilon}'(q_i)}{L}$.

### 5.1.1 Two-point dynamical correlation functions in the ordered phase

Following through the same steps as in the quench case we arrive at

$$
\chi^{xx}(\ell,t) = C \exp\left( \frac{1}{2\pi} \int_{-\pi}^{\pi} |t\varepsilon'(x) - \ell| \log(1 - 4\pi\rho(x)) dx \right),
\tag{143}
$$

where, using the results of Section 3,

$$
C = \begin{cases} \xi \exp\left(-2\int_{-\pi}^{\pi}\int_{-\pi}^{\pi} \frac{\rho(y)\rho'(x)}{\tan\left(\frac{x-y}{2}\right)} dx dy \right) & \text{in the space like-region at order } \rho_\beta^2 \\ \xi & \text{in the time like-region at order } \rho_\beta^0. \end{cases}
\tag{144}
$$

As far as we know the result (143) has not previously been obtained in the literature.

### 5.1.2 Numerical checks

In order to check the accuracy of (143) at finite times we have carried out numerical simulations following Ref. [92], where the finite temperature dynamical two-point function for a finite open chain is computed exactly as a Pfaffian of a known matrix. As long as the two points are sufficiently far from the boundaries, then they take almost the same values as in an infinite chain. In Fig. 11 we compare (143) to numerical results in the space-like region by considering the logarithm of the correlator

$$
\mathcal{L}(\ell,t) = \log\left( \langle \sigma_{\ell+1}^x(t)\sigma_1^x(0) \rangle \right).
\tag{145}
$$

For simplicity we take the extreme case $\alpha = 0$, which corresponds to setting $t = 0$. We recall indeed that static correlations are also covered by our calculations, as explained in Section 3.5.3. In the left panel we plot (143) as a function of distance $\ell$ and mark by red crosses numerical results obtained for a chain of $L = 200$ sites. We see that the asymptotic result (143) is in excellent agreement for $\alpha = 0$. In the right panel we test the accuracy of the $\mathcal{O}(\rho_\beta^2)$ value of the prefactor (144) by considering the quantity

$$
\Gamma(\beta) = \frac{\xi - \left\langle \sigma_{\ell+1}^x(0)\,\sigma_1^x(0) \right\rangle e^{\ell \int_{-\pi}^{\pi} \frac{dx}{2\pi} \log(1-4\pi\rho(x))}}{\xi - C},
\tag{146}
$$

where $C$ is given by (144).

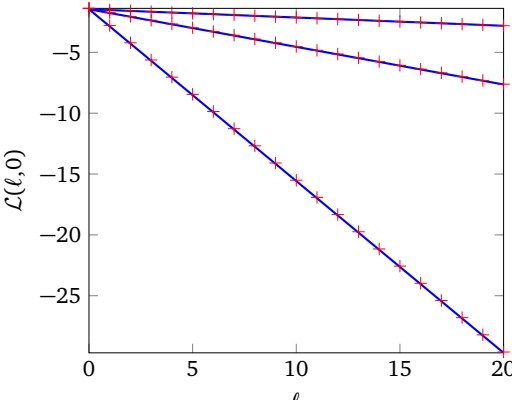
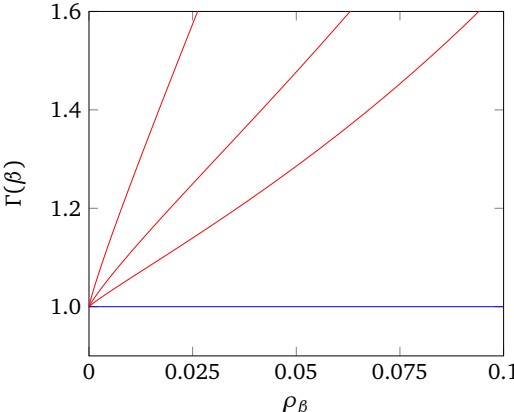

Figure 11: Left: $\mathcal{L}(\ell, 0)$ for $h = 0.5$ and $\beta = 2, 1, 0.25$ (top to bottom). Numerical results for a $L = 200$ site open chain are shown as red crosses and equation (143) by a continuous blue line. Right: Numerically determined $\Gamma(\beta)$ as a function of $\rho_\beta$ for $\ell = 30$ and $h = 3/4, 1/2, 1/4$ from top to bottom; in blue is the expected value as $\rho_\beta \to 0$.

We see that $\Gamma(\beta)$ approaches 1 for small values of $\rho_\beta$, which means that the prefactor (144) is indeed correct to order $\mathcal{O}(\rho_\beta^2)$. The linear increase in $\rho_\beta$ shows that the correction to our result for $C$ is $\mathcal{O}(\rho_\beta^3)$.

We now turn to the time-like region. In Fig. 12 we compare numerical results for $\mathcal{L}(\ell, t)$ as a function of $t$ for different values of $\ell$ to (143). We recall that in the time-like region equation (143) only gives the leading time behaviour, i.e. the exponent of the exponential decay. Hence only the slope of our analytic result for $\mathcal{L}(\ell, t)$ has to match the numerics. This is indeed seen to be the case in Fig. 12. In the space-like regime, i.e. at sufficiently short times, (143) is again seen to be in very good agreement with the numerical results.

In order to have a more quantitative check on the exponential factor in (143) it is useful to consider the difference

$$\mathcal{L}(\ell, t) - \log \chi^{xx}(\ell, t), \tag{147}$$

where $\chi^{xx}(\ell, t)$ is given by (143). If our exponential factor is exact, the difference should decay at late times more slowly than exponential, i.e. supposedly as a power law. In Figure 13 we plot (147) as a function of $\log Jt$ and indeed observe a linear behaviour. This confirms that the exponential factor in (143) is exact and moreover establishes that the subleading asymptotics is a power law in time.

## 5.2 Disordered phase $h > 1$

### 5.2.1 An approximate mapping to the ordered case

In the disordered phase the form factors in (36) still differ from the finite temperature ordered case by the fact that there are no intermediate states with the same number of momenta as in the representative state. From Section 3.2 we know that the terms $M = N + 1$ and $M = N - 1$ are equally important at late times, but the subleading terms are dominant in $\rho_\beta$ for $M = N + 1$. Since we are interested in the leading time behaviour only, with the order $\rho_\beta^0$ subleading corrections, we will restrict our analysis to the case $M = N + 1$.

In this case the partial fraction decomposition (73) of $|_{\text{NS}}\langle q_1, ..., q_N | \sigma_l^x | p_1, ..., p_M \rangle_R|^2$ necessarily involves one $\nu_j = 0$. To obtain the $p_j$ dependence of the corresponding $A$'s in (73), it suffices to decompose the form factor $|_{\text{NS}}\langle q_1, ..., q_N | \sigma_l^x | p_1, ..., p_M \rangle_R|^2$ by starting to write the

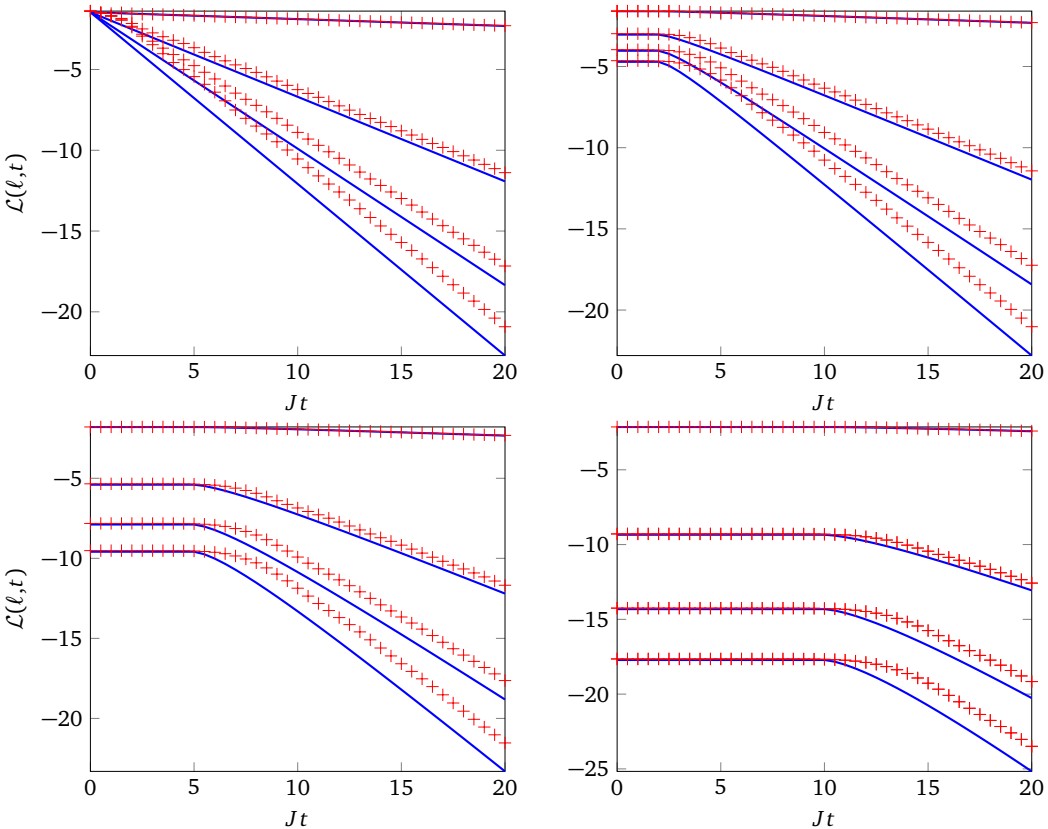

Figure 12: $\mathcal{L}(\ell, t)$ as a function of $t$ for $h = 0.5$ and $\beta = 2, 0.5, 0.286, 0.2$ from top to bottom inside each panel, with $\ell = 0, 2, 5, 10$ from top left to bottom right. Numerical results for a $L = 200$ site open chain are shown by red crosses and equation (143) by a straight blue line. The slopes of the numerical results are correctly reproduced by (143) in the time-like region, i.e. for $v_{\max} t > \ell$.

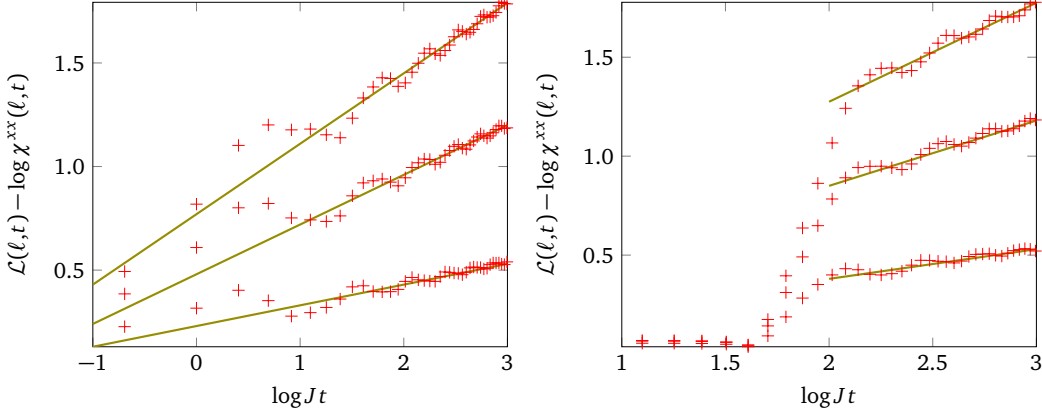

Figure 13: $\mathcal{L}(\ell, t) - \log \chi^{xx}(\ell, t)$ with $\chi^{xx}(\ell, t)$ given by (143) as a function of $\log J t$ for $h = 0.5$ and $\beta = 0.5, 0.286, 0.2$ from bottom to top inside each panel, with $\ell = 0$ (left) and $\ell = 5$ (right) in red. In green is indicated a linear fit in the time-like region $v_{\max} t > \ell$.

partial fraction decomposition with respect to $p_j$ and retaining only the $v_j = 0$ part (which corresponds to $P_0(X)$ in (39)), and decomposing with respect to the other momenta in the

usual fashion. This part is $\frac{\prod_{i=1}^{N} \varepsilon_{p_j q_i}^2}{\prod_{i=1}^{N+1} \varepsilon_{p_j p_i}}$, that is $\frac{1}{\varepsilon(p_j)}$ times $\frac{\prod_{i=1}^{N} \varepsilon_{p_j q_i}^2}{\prod_{i=1\neq j}^{N+1} \varepsilon_{p_j p_i}}$. This latter factor satisfies the hypotheses of (112) which allows us to replace it by 1 as long as we are interested only in the leading time behaviour. It follows that under this approximation the $v_j = 0$ terms in the partial fraction decomposition (73) of $|_{\text{NS}}\langle q_1, ..., q_N | \sigma_\ell^x | p_1, ..., p_M \rangle_{\text{R}}|^2$ contribute to (36) as

$$
\frac{2J\sqrt{h}\xi}{(N+1)!L} \sum_{p_j \in \text{R}} \frac{e^{-it\overline{\varepsilon}(p_j)}}{\varepsilon(p_j)} \sum_{p_i \in \text{R}, i \neq j} |_{\text{R}}\langle p_1, ..., p_{j-1} p_{j+1}, ..., p_{N+1} | \sigma_\ell^x | q_1, ..., q_N \rangle_{\text{NS}}|^2
$$
$$
\times e^{it[E(\{q\})-E(\{p\}-\{p_j\})]+i\ell[P(\{p\}-\{p_j\})-P(\{q\})]} \,. \tag{148}
$$

Taking into account all the possible $j = 1, ..., N+1$ for which one can have $v_j = 0$ in (73), we obtain

$$
\chi^{xx}(\ell, t) \approx \left( \frac{2J\sqrt{h}\xi}{L} \sum_{p \in \text{R}} \frac{e^{-it\overline{\varepsilon}(p)}}{\varepsilon(p)} \right)
$$
$$
\times \frac{1}{N!} \sum_{\substack{p_1, ..., p_N \\ \in \text{R}}} |_{\text{R}}\langle p_1, ..., p_N | \sigma_\ell^x | q_1, ..., q_N \rangle_{\text{NS}}|^2 e^{it(E(\{q\})-E(\{p\}))+i\ell(P(\{p\})-P(\{q\}))} \,, \tag{149}
$$

where $\{p\} = \{p_1, ..., p_N\}$ has now the same number of momenta as in the representative state. The second factor in (149) is precisely of the same form as the one we considered in the ordered phase.

### 5.2.2 Two-point dynamical correlation functions in the disordered phase

Putting everything together we obtain the following result for the leading late time behaviour of the two-point function

$$
\chi^{xx}(\ell, t) \approx C(\ell, t) \exp\left( \frac{1}{2\pi} \int_{-\pi}^{\pi} |t\varepsilon'(x) + \ell| \log(1 - 4\pi\rho(x)) dx \right) , \tag{150}
$$

where (cf. section 3.2)

$$
C(\ell, t) = \begin{cases} 2J\sqrt{h}\xi \int_{-\pi}^{\pi} \left( \frac{e^{-it\overline{\varepsilon}(x)}}{2\pi} + 2\rho(x)\cos(t\overline{\varepsilon}(x)) \right) \frac{dx}{\varepsilon(x)} & \text{if } v_{\max} t < \ell \text{ at } \mathcal{O}(\rho_\beta) \\ 2J\sqrt{h}\xi \int_{-\pi}^{\pi} \frac{e^{-it\overline{\varepsilon}(x)}}{2\pi\varepsilon(x)} dx & \text{if } v_{\max} t > \ell \text{ at } \mathcal{O}(\rho_\beta^0) , \end{cases} \tag{151}
$$

$\ell, t \geq 0$, and $v_{\max}$ is defined in (38).

### 5.2.3 Numerical checks

We have checked the accuracy of (150) by comparing it to numerical calculations following Ref. [92]. In the following we show results for

$$
\mathcal{R}(\ell, t) = \text{Re}\left( \langle \sigma_{\ell+1}^x(t) \sigma_1^x(0) \rangle \right) . \tag{152}
$$

In the space-like region $v_{\max} t < \ell$ we furthermore check our result for the prefactor $C(\ell, t)$ (151) by computing

$$
\Lambda(\ell, \beta) = \frac{\mathcal{R}(\ell, 0) e^{-\ell \int_{-\pi}^{\pi} \frac{dx}{2\pi} \log(1 - 4\pi\rho(x))} - C_0(\ell, 0)}{C(\ell, 0) - C_0(\ell, 0)} , \tag{153}
$$

where $C_0(\ell, t) = \frac{J\sqrt{h}\xi}{\pi} \int_{-\pi}^{\pi} \frac{dx}{\varepsilon(x)} e^{-it\overline{\varepsilon}(x)}$ is the $\mathcal{O}(\rho_\beta^0)$ contribution to $C(\ell, t)$. If and only if the prefactor in (151) is correct at order $\mathcal{O}(\rho_\beta)$, $\Lambda(\ell, \beta) \to 1$ when $\beta \to 0$.

In Fig. 14 we compare our analytic expression (150) in the space-like region to numerical results for $R(\ell, t)$.

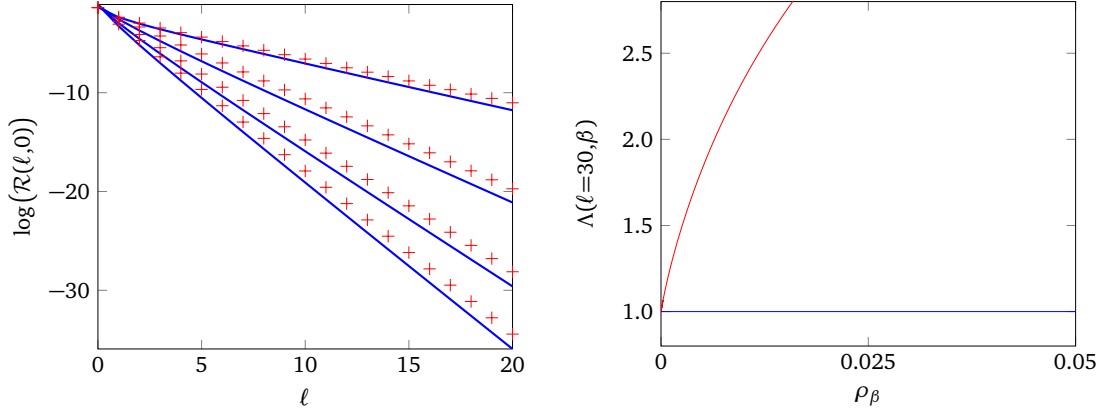

Figure 14: Left: $\log\big(\mathcal{R}(\ell, 0)\big)$ as a function of $\ell$ for $h = 1.5$ and $\beta = 2, 0.5, 0.285, 0.2$ from top to bottom. Numerical results for a $L = 200$ site open chain are shown as red crosses and equation (150) as straight blue lines. Right: numerical results for $\Lambda(\ell = 30, \beta)$ as a function of $\rho_\beta$ for $h = 3/2$ (red line). The expected result when $\rho_\beta = 0$ is shown in blue.

Our analytic expression is seen to be in good agreement with the numerical results and the remaining discrepancy is due to $\mathcal{O}(\rho_\beta^2)$ corrections to the prefactor.

In Fig. 15 we present results for $\mathcal{R}(\ell, t)$ in the time-like region $v_{\max} t > \ell$ at low temperatures. We see that the analytical result is in excellent agreement with the numerics.

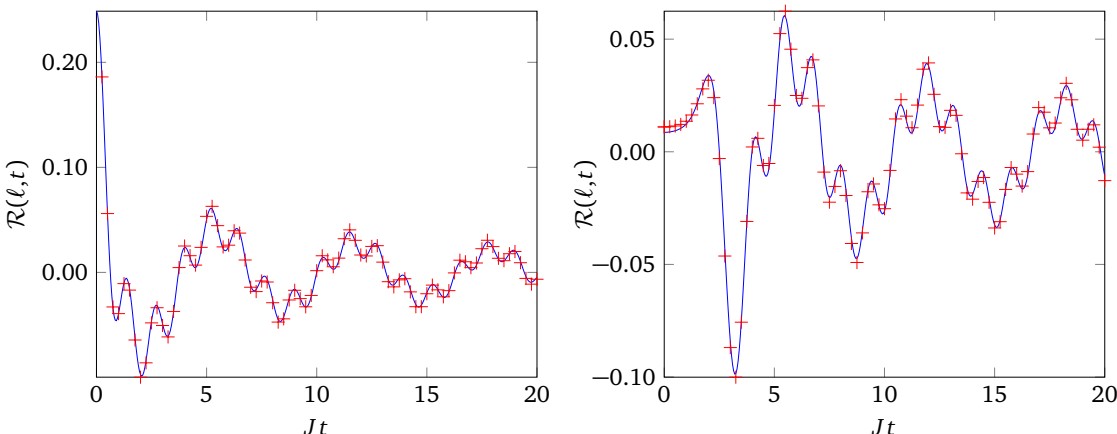

Figure 15: $\mathcal{R}(\ell, t)$ as a function of $t$ with $h = 1.5$ and $\beta = 3$ for $\ell = 0$ (left) and $\ell = 5$ (right). Numerical results for a $L = 200$ site open chain are shown as red crosses and equation (150) as a solid blue line.

In order to check the accuracy of our result for the exponential decay of the two-point function for intermediate and high temperatures we compare (150) to numerical results $\log |\mathcal{R}(\ell, t)|$ in Fig. 16. As our result for the prefactor $C(\ell, t)$ only holds at low temperatures we expect the numerical results to differ from our analytical prediction by an essentially constant offset. This is indeed what we observe.

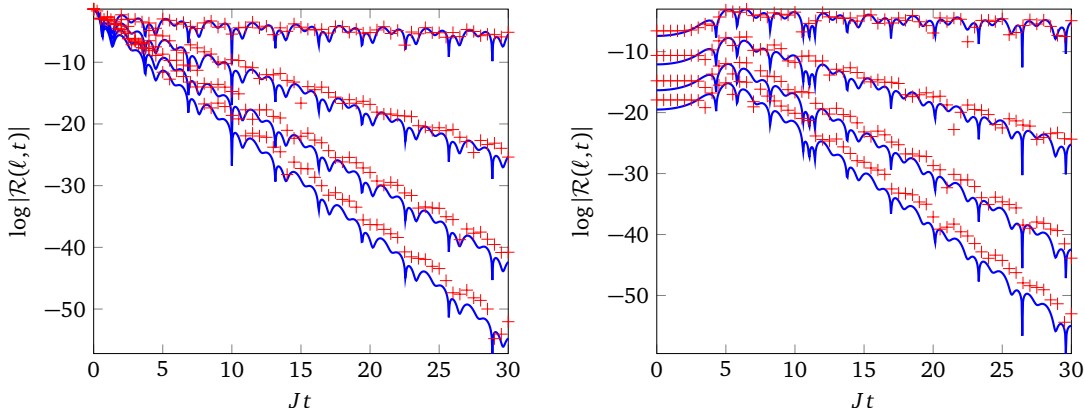

Figure 16: $\log|\mathcal{R}(\ell, t)|$ as a function of $t$ for $h = 1.5$ and $\beta = 2, 0.5, 0.286, 0.2$ from top to bottom inside each panel, with $\ell = 0$ (left) and $\ell = 10$ (right). Numerical results for a $L = 200$ site open chain are shown as red crosses and equation (150) as a solid blue line.

## 6 Summary and Discussion

In this work we have considered two problems in the transverse field Ising model: (i) The time dependence of the order parameter after a quantum quench; (ii) The dynamical order parameter two point function in equilibrium at finite temperatures. Using the quench action approach for (i) and a micro-canonical formulation combined with typicality ideas for (ii) these problems can both be formulated in terms of spectral representations using Hamiltonian eigenstates, i.e. sums over form factors.

These highly intricate sums over a macroscopic quantity of momenta of a form factor with many singularities represent however a considerable technical challenge. We showed that in the case of semi-local operators such as $\sigma_j^x$ in the TFIM, this difficulty can be addressed by decomposing the form factor into partial fractions, which permits the sums over momenta to be decoupled and the late time behaviour to be determined. These partial fractions can be organized in terms of the degree and the position of their poles, which naturally leads to an expansion in the density of particles of the representative state. The leading behaviour at late times can be then computed at all orders in the density through a determinant representation of these poles, which leads invariably to an exponential decay.

Our analysis provides a precise characterization of the excitations over the representative state that contributes to the late time behaviour of correlation functions of semi-local operators. We find that *simultaneous* particle-hole excitations of *all* particles in the representative state contribute to the correlation function, and we altogether have to sum over $\mathcal{O}\big((\epsilon L)^N\big)$ excited states, where $\epsilon$ is a fixed number that can be taken as small as desired and $N/L$ is the density. In particular, this implies that the appealing picture of an expansion in terms of a finite number of particle-hole excitations over the full momentum space *fails* for semi-local operators at finite temperatures. The form factor sum is dominated by mesoscopic excitations (in the sense given in Section 3.5.1) around each single particle, which is an exponential number of states, but subentropic in the sense that includes only states whose macroscopic state is the representative state itself.

We have compared our analytic results to numerical computations in both the finite temperature and quench contexts. In the absence of saddle points, thus where the sums (46) are valid at all momenta, we find remarkable agreement at all times. In cases where saddle points

occur our numerical results indicate the presence of a multiplicative power-law behaviour as a subleading correction to the exponential decay. We believe that this will emerge from saddle point effects of these classes of excitations, due to the fact that (46) is not valid anymore close to the saddle point. Higher-order corrections in time will also arise from contributions with $v_i = 0$ in (42); in this case there is no singularity and the full momentum space should be summed over to take them into account. We believe that partial fraction decompositions for form factors of semi-local operators are not only suited for extracting the late time asymptotics, but should also be useful for determining intermediate and short time behaviours.

In interacting models, the singularity structure of the form factors of semi-local operators is not fundamentally changed and a partial fraction decomposition will provide a useful organizing principle as well. A notable difference is that the Bethe equations link the different particles so that the momenta sums can never fully decouple. This will be the subject of a subsequent paper.

For local operators however, the story is radically different. The singularity structure of the form factors (12) is completely dissimilar the states that dominate the late time asymptotics are selected by different principles. Excitations over the full momentum space and not only near the singularities will play equally important roles. This will be discussed elsewhere.

# Acknowledgements

We are grateful to Jacopo de Nardis, Frank Göhmann and Karol Kozlowski for very helpful discussions and comments. This work was supported by the EPSRC under grant EP/S020527/1 (FHLE and EG) and by the ERC under Starting Grant No. 805252 LoCoMacro (MF).

# A  Riemann sums of singular functions

Sums of form factors lead to Riemann sums of functions with a quadratic singularity of the form

$$\frac{1}{L} \sum_{k=-L+1}^{L} f\left(\tfrac{k}{L}\right) \qquad \text{with } f(x) \underset{x\to 0}{\sim} \frac{1}{x^2} . \tag{154}$$

Since the integral of $f(x)$ is divergent, the limit $L \to \infty$ cannot be directly taken as for regular functions and needs special treatment. The purpose of this appendix is to explain techniques to compute them. We note that an alternative way of treating such sums is to employ contour integral techniques [93].

## A.1  One-dimensional sums

Oscillatory Riemann sums of functions with a quadratic singularity cannot be estimated with the usual results on stationary phase approximation to obtain their large time behaviour. The principle is then to add and remove an elementary function with the same singularity, but whose Riemann sum can be computed directly, so that the remaining function has no singularity and thus has an oscillatory Riemann sum that vanishes at large times.

### A.1.1  Generic example

For concreteness, let us illustrate this procedure with the Riemann sum

$$S_L(\theta) = \frac{1}{L} \sum_{k=-L+1\neq 0}^{L} f\left(\tfrac{k}{L}\right) e^{i\frac{k}{L}\theta} , \tag{155}$$

for $f(x)$ a function such that $f(x) = \frac{1}{x^2} + \mathcal{O}(x^0)$ for $x \to 0$, $f$ being regular otherwise. Then

$$S_L(\theta) = \frac{1}{L} \sum_{k=-L+1\neq 0}^{L} \left(\tfrac{L}{k}\right)^2 e^{i\frac{k}{L}\theta} + \frac{1}{L} \sum_{k=-L+1}^{L} \bar{f}\left(\tfrac{k}{L}\right) e^{i\frac{k}{L}\theta}, \tag{156}$$

with $\bar{f}(x) = f(x) - 1/x^2$ a regular function. The second term on the right-hand side can be turned into an integral, while we rewrite the first as follows

$$\frac{1}{L} \sum_{k=-L+1\neq 0}^{L} \left(\tfrac{L}{k}\right)^2 e^{i\frac{k}{L}\theta} = L \sum_{k=-\infty,\neq 0}^{+\infty} \frac{1}{k^2} e^{i\frac{k}{L}\theta} - \frac{1}{L} \sum_{k=L+1}^{+\infty} \left(\tfrac{L}{k}\right)^2 e^{i\frac{k}{L}\theta} - \frac{1}{L} \sum_{k=-\infty}^{-L} \left(\tfrac{L}{k}\right)^2 e^{i\frac{k}{L}\theta}. \tag{157}$$

The second and third Riemann sums can now be turned into integrals without any problems, which gives

$$S_L(\theta) = L \sum_{k=-\infty,\neq 0}^{+\infty} \frac{e^{i\frac{k}{L}\theta}}{k^2} + \int_{-1}^{1} \bar{f}(x) e^{i\theta x} dx - \int_{1}^{\infty} \frac{e^{i\theta x}}{x^2} dx - \int_{-\infty}^{-1} \frac{e^{i\theta x}}{x^2} dx + \mathcal{O}(L^{-1}). \tag{158}$$

The three integrals are oscillatory integrals of bounded functions and hence vanish for large $\theta$. We conclude that

$$S_L(\theta) = L \sum_{k=-\infty,\neq 0}^{+\infty} \frac{e^{i\frac{k}{L}\theta}}{k^2} + \mathcal{O}(L^0 \theta^{-1/2}) + \mathcal{O}(L^{-1}). \tag{159}$$

### A.1.2 Elementary oscillatory sums with singularities

The above analysis shows that the leading asymptotics of Riemann sums with simple or double poles involves the following sums

$$\sum_{n\in\mathbb{Z}} \frac{e^{iw(n+\alpha)}}{(n+\alpha)} = \frac{\pi}{\sin\pi\alpha} e^{i\pi\alpha\,\mathrm{sgn}(w)}, \qquad \text{for } -\pi < w \le \pi \tag{160}$$

$$\sum_{n\in\mathbb{Z}} \frac{e^{iw(n+\alpha)}}{(n+\alpha)^2} = \left(\frac{\pi}{\sin\pi\alpha}\right)^2 + \frac{i\pi}{\sin\pi\alpha} w e^{i\pi\alpha\,\mathrm{sgn}(w)}, \qquad \text{for } -\pi < w \le \pi.$$

These are readily obtained by computing the Fourier series coefficients of the right-hand sides multiplied by $e^{-iw\alpha}$ and seen as a $2\pi$-periodic function of $w$. In particular we have

$$\sum_{n\in\mathbb{Z}} \frac{e^{iw(n+1/2)}}{(n+1/2)^m} = \begin{cases} \pi^2\left(1 - \frac{|w|}{\pi}\right) & \text{if } m = 2, \\ i\pi\,\mathrm{sgn}(w) & \text{if } m = 1. \end{cases} \tag{161}$$

$$\sum_{n\in\mathbb{Z}\setminus\{0\}} \frac{e^{iwn}}{n^m} = \begin{cases} \pi^2\left(\frac{1}{3} - \frac{|w|}{\pi} + \frac{w^2}{2\pi^2}\right) & \text{if } m = 2, \\ i(\pi - |w|)\,\mathrm{sgn}(w) & \text{if } m = 1. \end{cases}$$

### A.1.3 Sums arising in finite-temperature dynamics

Following the steps outlined above we obtain for $q \in \mathrm{NS}$

$$\sum_{p\in\mathrm{R}} \frac{e^{-it\bar{\varepsilon}(p)}}{L\sin^2\left(\frac{p-q}{2}\right)} = L \sum_{k=-\infty}^{+\infty} \frac{e^{-it\bar{\varepsilon}(q+2\pi(k+1/2)/L)}}{\pi^2(k+1/2)^2} - \int_{\pi}^{\infty} \frac{4e^{-it\bar{\varepsilon}(x)}}{(x-q)^2} \frac{dx}{2\pi} - \int_{-\infty}^{-\pi} \frac{4e^{-it\bar{\varepsilon}(x)}}{(x-q)^2} \frac{dx}{2\pi}$$

$$+ \int_{-\pi}^{\pi} e^{-it\bar{\varepsilon}(x)} \left(\frac{1}{\sin^2\frac{x-q}{2}} - \frac{1}{(x-q)^2/4}\right) \frac{dx}{2\pi} + \mathcal{O}(L^{-1}). \tag{162}$$

At leading order in time, one can Taylor expand the $\overline{\varepsilon}$ in the remaining sum to fall back on an elementary oscillatory sum. The three integrals involve oscillatory bounded functions and hence decay to zero with time.

Similarly we obtain for $q \in \text{NS}$

$$
\sum_{p \in \text{R}} \frac{e^{-it\overline{\varepsilon}(p)}}{L \sin\left(\frac{p-q}{2}\right)} = \sum_{k=-\infty}^{+\infty} \frac{e^{-it\overline{\varepsilon}(q+2\pi(k+1/2)/L)}}{\pi(k+1/2)} - \int_{\pi}^{\infty} \frac{2e^{-it\overline{\varepsilon}(x)}}{(x-q)} \frac{dx}{2\pi} - \int_{-\infty}^{-\pi} \frac{2e^{-it\overline{\varepsilon}(x)}}{(x-q)} \frac{dx}{2\pi}
$$
$$
+ \int_{-\pi}^{\pi} e^{-it\overline{\varepsilon}(x)} \left( \frac{1}{\sin\frac{x-q}{2}} - \frac{1}{(x-q)/2} \right) \frac{dx}{2\pi} + \mathcal{O}\left(L^{-1}\right),
\tag{163}
$$

where the integrals are again integrals of oscillatory bounded functions and vanish at late times.

Finally we use (161) to obtain the asymptotic values of the sums in (162) and (163) for $q \in \text{NS}$ and $\overline{\varepsilon}'(q) \neq 0$

$$
\sum_{p \in \text{R}} \frac{e^{-it\overline{\varepsilon}(p)}}{L \sin\left(\frac{p-q}{2}\right)} = -i\,\text{sgn}(t\overline{\varepsilon}'(q))e^{-it\overline{\varepsilon}(q)} + \mathcal{O}\left(L^0 t^{-1/2}\right) + \mathcal{O}\left(L^{-1}\right),
$$
$$
\sum_{p \in \text{R}} \frac{e^{-it\overline{\varepsilon}(p)}}{L^2 \sin^2\left(\frac{p-q}{2}\right)} = \left(1 - \frac{2\left|t\overline{\varepsilon}'(q)\right|}{L}\right) e^{-it\overline{\varepsilon}(q)} + \mathcal{O}\left(L^{-1}t^{-1/2}\right) + \mathcal{O}\left(L^{-2}\right).
\tag{164}
$$

### A.1.4 Sums arising in quantum quench dynamics (90)

We now turn to momentum sums of the form

$$
\Sigma^{(n)}(q,q',t) = \frac{4}{L^n} \sum_{0<p\in\text{R}} \frac{\sin p \sin q' f(p)}{(\cos q - \cos p)^n f(q')} e^{2it(\varepsilon(p)-\varepsilon(q))}, \quad n=1,2,
\tag{165}
$$

where $f(q)$ is defined in (87). Using that $\frac{\sin x f(x)}{\cos q - \cos x} - \frac{f(q)}{x-q}$ is a bounded function of $x$ we can proceed along the same lines as in Section A.1.3 to conclude that for $q \in \text{NS}$

$$
\Sigma^{(1)}(q,q',t) = \frac{4}{L} \sum_{0<p\in\text{R}} \frac{\sin q'}{p-q} e^{2it(\varepsilon(p)-\varepsilon(q))} \frac{f(q)}{f(q')} + \mathcal{O}(L^0 t^{-1/2}).
\tag{166}
$$

At leading order in time we may then Taylor expand $\varepsilon(p)$ around $p = q$, and write $p = q + \frac{2\pi}{L}(n+1/2)$ to fall back on one of the oscillatory sums in (161). In this way we obtain our final result

$$
\Sigma^{(1)}(q,q',t) = 2i\,\text{sgn}(t\varepsilon'(q))\sin q' \frac{f(q)}{f(q')} + \mathcal{O}(L^0 t^{-1/2}).
\tag{167}
$$

The analysis of the $\Sigma^{(2)}(q,q',t)$ proceeds in complete analogy: we use that $\frac{\sin x f(x)}{(\cos x - \cos q)^2} - \frac{f(q)}{\sin q(x-q)^2} - \frac{f'(q)}{\sin q(x-q)}$ is a bounded function of $x$ to conclude that

$$
\Sigma^{(2)}(q,q',t) = \frac{4}{L^2} \sum_{p>0,\in\text{R}} \frac{e^{2it(\varepsilon(p)-\varepsilon(q))}}{(p-q)^2} + \frac{4}{L^2} \sum_{p>0,\in\text{R}} \frac{f'(q)}{f(q)} \frac{e^{2it(\varepsilon(p)-\varepsilon(q))}}{p-q} + \mathcal{O}\left(L^{-1}t^{-1/2}\right).
\tag{168}
$$

Taylor expanding $\varepsilon(p)$ around $p = q$ and using (161) we finally arrive at

$$
\Sigma^{(2)}(q,q',t) = 1 - \frac{4|t\varepsilon'(q)|}{L} + \frac{2i\,\text{sgn}(t\varepsilon'(q))}{L} \frac{f'(q)}{f(q)} + \mathcal{O}(L^{-1}t^{-1/2}).
\tag{169}
$$

### A.2 Two-dimensional sums

In this subsection we calculate the two-dimensional sums arising in Sections 3.1.5 and 3.4. Apart from being two-dimensional, the sums treated in this section differ from the previous section by the fact that they are performed over the particles of the representative state, and not over arbitrary, regularly spaced momenta in the Ramond sector.

#### A.2.1 Sums arising in finite-temperature dynamics

We consider

$$\Omega_1(t) = \frac{1}{L^2} \sum_{i \neq j} \frac{e^{it(\overline{\varepsilon}(q_j) - \overline{\varepsilon}(q_i))}}{\sin^2\left(\frac{q_i - q_j}{2}\right)} \operatorname{sgn}\left(\overline{\varepsilon}'(q_j)\overline{\varepsilon}'(q_i)\right). \tag{170}$$

This sum is divergent when $L \to \infty$. It grows as $\propto L$ and the proportionality constant depends on the realization of the root density $\rho$ in finite-size through the $q_i$'s, and as a consequence cannot be written in terms of the root density. However, this prefactor does not depend on $t$, and the difference $\Omega_1(t) - \Omega_1(0)$ which appears in the main text is not divergent in the thermodynamic limit.

We have by symmetrizing the sum over $i, j$

$$\Omega_1(t) - \Omega_1(0) = \frac{1}{2L^2} \sum_{i \neq j} \frac{e^{it(\overline{\varepsilon}(q_j) - \overline{\varepsilon}(q_i))} + e^{it(\overline{\varepsilon}(q_i) - \overline{\varepsilon}(q_j))} - 2}{\sin^2\left(\frac{q_i - q_j}{2}\right)} \operatorname{sgn}\left(\overline{\varepsilon}'(q_j)\overline{\varepsilon}'(q_i)\right). \tag{171}$$

The summand does not have poles anymore, so that the sum can be turned into an integral in the $L \to \infty$ limit

$$\Omega_1(t) - \Omega_1(0) =$$
$$\frac{1}{2} \int_{-\pi}^{\pi} \int_{-\pi}^{\pi} \frac{e^{it(\overline{\varepsilon}(v) - \overline{\varepsilon}(u))} + e^{it(\overline{\varepsilon}(u) - \overline{\varepsilon}(v))} - 2}{\sin^2\left(\frac{u - v}{2}\right)} \operatorname{sgn}\left(\overline{\varepsilon}'(u)\overline{\varepsilon}'(v)\right) \rho(u)\rho(v) du dv + \mathcal{O}(L^{-1}). \tag{172}$$

We now have to determine the large $t$ behaviour of this expression. We first write

$$\Omega_1(t) - \Omega_1(0) = -2 \int_{-\pi}^{\pi} \int_{-\pi}^{\pi} \frac{\sin^2[t(\overline{\varepsilon}(v) - \overline{\varepsilon}(u))/2]}{\sin^2\left(\frac{u-v}{2}\right)} \operatorname{sgn}\left(\overline{\varepsilon}'(u)\overline{\varepsilon}'(v)\right) \rho(u)\rho(v) du dv, \tag{173}$$

and perform a change of variable $v = u + \eta/t$

$$\Omega_1(t) - \Omega_1(0) =$$
$$-2 \int_{-\pi}^{\pi} du \int_{(-\pi-u)|t|}^{(\pi-u)|t|} d\eta \frac{\sin^2[t(\overline{\varepsilon}(u + \frac{\eta}{t}) - \overline{\varepsilon}(u))/2]}{|t| \sin^2\left(\frac{\eta}{2t}\right)} \operatorname{sgn}\left(\overline{\varepsilon}'(u)\overline{\varepsilon}'(u + \frac{\eta}{t})\right) \rho(u)\rho(u + \frac{\eta}{t}). \tag{174}$$

At leading order in $t$, it yields

$$\Omega_1(t) - \Omega_1(0) = -8|t| \int_{-\pi}^{\pi} du \int_{-\infty}^{\infty} d\eta \frac{\sin^2[\overline{\varepsilon}'(u)\eta/2]}{\eta^2} \rho(u)^2 + o(|t|). \tag{175}$$

Using $\int_{-\infty}^{\infty} \frac{\sin^2 x}{x^2} dx = \pi$, we obtain

$$\Omega_1(t) - \Omega_1(0) = -4\pi \int_{-\pi}^{\pi} |t\overline{\varepsilon}'(u)| \rho(u)^2 + o(|t|). \tag{176}$$

Let us determine the sub-leading term $o(|t|)$ in the space-like regime, i.e. when $\text{sgn}(\overline{\varepsilon}'(u))$ is constant, and when the root density $\rho$ is continuous. In this case, coming back to (172), we integrate by part to obtain

$$
\begin{aligned}
\Omega_1(t) - \Omega_1(0) = {} & 2t \int_{-\pi}^{\pi} \int_{-\pi}^{\pi} \overline{\varepsilon}'(v) \frac{\sin[t(\overline{\varepsilon}(v) - \overline{\varepsilon}(u))]}{\tan\left(\frac{u-v}{2}\right)} \rho(u)\rho(v) du\, dv \\
& + 2 \int_{-\pi}^{\pi} \int_{-\pi}^{\pi} \frac{1}{\tan\left(\frac{u-v}{2}\right)} \rho(u)\rho'(v) du\, dv \\
& - \int_{-\pi}^{\pi} \int_{-\pi}^{\pi} \frac{e^{it(\overline{\varepsilon}(v) - \overline{\varepsilon}(u))} + e^{it(\overline{\varepsilon}(u) - \overline{\varepsilon}(v))}}{\tan\left(\frac{u-v}{2}\right)} \rho(u)\rho'(v) du\, dv\,,
\end{aligned}
\tag{177}
$$

where the last two double integrals are understood in principal value.

Let us focus first on the last double integral, that we separate into two integration regions, one with $|u - v| > \epsilon$ and one with $|u - v| < \epsilon$ for a small fixed $\epsilon > 0$. In the first region, the term is an oscillatory integral of a bounded function, hence decays to zero with time. In the second region, the $v$ integral at fixed $u$ may be approximated by

$$
\approx 2\rho'(u) \int_{-f_-(u)}^{f_+(u)} \frac{e^{itx\overline{\varepsilon}'(u)} + e^{-itx\overline{\varepsilon}'(u)}}{x} dx\,,
\tag{178}
$$

with $0 < f_\pm(u) < \epsilon$, where $f_\pm(u)$ are some limits that depend on $u$. Assuming without loss of generality $f_-(u) < f_+(u)$, this integral is $\int_{f_-(u)t}^{f_+(u)t} \frac{e^{iy\overline{\varepsilon}'(u)}}{y} dy - \int_{-f_+(u)t}^{-f_-(u)t} \frac{e^{iy\overline{\varepsilon}'(u)}}{y} dy$, which decays to zero when $t \to \infty$. Hence

$$
\begin{aligned}
\Omega_1(t) - \Omega_1(0) = {} & 2t \int_{-\pi}^{\pi} \int_{-\pi}^{\pi} \overline{\varepsilon}'(v) \frac{\sin[t(\overline{\varepsilon}(v) - \overline{\varepsilon}(u))]}{\tan\left(\frac{u-v}{2}\right)} \rho(u)\rho(v) du\, dv \\
& + 2 \int_{-\pi}^{\pi} \int_{-\pi}^{\pi} \frac{1}{\tan\left(\frac{u-v}{2}\right)} \rho(u)\rho'(v) du\, dv \\
& + o(t^0)\,.
\end{aligned}
\tag{179}
$$

We now focus on the first double integral. First, since in the space-like regime $\overline{\varepsilon}'(u)$ never vanishes, $\overline{\varepsilon}(u)$ is one-to-one from $[-\pi, \pi]$ to $[\varepsilon_{\min}, \varepsilon_{\max}]$. Hence one can perform a change of variable $x = \overline{\varepsilon}(u), y = \overline{\varepsilon}(v)$ to obtain

$$
\begin{aligned}
& \int_{-\pi}^{\pi} \int_{-\pi}^{\pi} \overline{\varepsilon}'(v) \frac{\sin[t(\overline{\varepsilon}(v) - \overline{\varepsilon}(u))]}{\tan\left(\frac{u-v}{2}\right)} \rho(u)\rho(v) du\, dv = \\
& \int_{\varepsilon_{\min}}^{\varepsilon_{\max}} \int_{\varepsilon_{\min}}^{\varepsilon_{\max}} \frac{1}{\overline{\varepsilon}'(\overline{\varepsilon}^{-1}(x))} \frac{\sin[t(y - x)]}{\tan\left(\frac{\overline{\varepsilon}^{-1}(x) - \overline{\varepsilon}^{-1}(y)}{2}\right)} \rho(\overline{\varepsilon}^{-1}(x))\rho(\overline{\varepsilon}^{-1}(y)) dx\, dy\,.
\end{aligned}
\tag{180}
$$

We now make the following observation. For any regular function $f(x, y)$, the integral

$$
\int_{\varepsilon_{\min}}^{\varepsilon_{\max}} \int_{\varepsilon_{\min}}^{\varepsilon_{\max}} \sin[t(y - x)] f(x, y) dx\, dy
\tag{181}
$$

is $o(t^{-1})$. Indeed, by integrating by part the $y$ integral we make appear a $1/t$, and the remaining integrals are oscillatory integrals, hence decay to zero with time. By adding and subtracting the appropriate term to the right-hand side of (180) so as to cancel the pole, we

conclude from this

$$\int_{\varepsilon_{\min}}^{\varepsilon_{\max}} \int_{\varepsilon_{\min}}^{\varepsilon_{\max}} \frac{1}{\overline{\varepsilon}'(\overline{\varepsilon}^{-1}(x))} \frac{\sin[t(y-x)]}{\tan\left(\frac{\overline{\varepsilon}^{-1}(x)-\overline{\varepsilon}^{-1}(y)}{2}\right)} \rho(\overline{\varepsilon}^{-1}(x))\rho(\overline{\varepsilon}^{-1}(y))dxdy$$

$$= 2\int_{\varepsilon_{\min}}^{\varepsilon_{\max}} \int_{\varepsilon_{\min}}^{\varepsilon_{\max}} \frac{\sin[t(y-x)]}{x-y}\rho(\overline{\varepsilon}^{-1}(x))^2 dxdy + o(t^{-1}). \tag{182}$$

We now split the $x$ integral into several pieces $[x_n, x_{x+1}]$ with $x_{n+1} - x_n$ small enough so that $\rho$ can be approximated to be constant on these pieces. Then we have

$$\int_{x_n}^{x_{n+1}} dx \int_{\varepsilon_{\min}}^{\varepsilon_{\max}} dy \frac{\sin[t(y-x)]}{x-y} = -\frac{1}{t}\int_{|t|(\varepsilon_{\max}-x_{n+1})}^{|t|(\varepsilon_{\max}-x_n)} \mathrm{Si}(u)du + \frac{1}{t}\int_{|t|(\varepsilon_{\min}-x_{n+1})}^{|t|(\varepsilon_{\min}-x_n)} \mathrm{Si}(u)du, \tag{183}$$

with $\mathrm{Si}(u) = \int_0^u \frac{\sin x}{x}dx$ the sinus integral. Using the expansion at large $u > 0$

$$\mathrm{Si}(u) = \frac{\pi}{2} - \frac{\cos u}{u} + \mathcal{O}(u^{-2}), \tag{184}$$

one obtains

$$\int_{x_n}^{x_{n+1}} dx \int_{\varepsilon_{\min}}^{\varepsilon_{\max}} dy \frac{\sin[t(y-x)]}{x-y} = -\pi \, \mathrm{sgn}(t)(x_{n+1} - x_n) + \mathcal{O}(t^{-2}). \tag{185}$$

Summing over the windows $[x_n, x_{n+1}]$ we obtain

$$\int_{\varepsilon_{\min}}^{\varepsilon_{\max}} \int_{\varepsilon_{\min}}^{\varepsilon_{\max}} \frac{\sin[t(y-x)]}{x-y}\rho(\overline{\varepsilon}^{-1}(x))^2 dxdy = -\pi \, \mathrm{sgn}(t) \int_{\varepsilon_{\min}}^{\varepsilon_{\max}} \rho(\overline{\varepsilon}^{-1}(x))^2 dx + \mathcal{O}(t^{-2}), \tag{186}$$

which yields in the space-like regime

$$\Omega_1(t) - \Omega_1(0) = -4\pi \int_{-\pi}^{\pi} |t\overline{\varepsilon}'(u)|\rho(u)^2 + 2\int_{-\pi}^{\pi}\int_{-\pi}^{\pi} \frac{1}{\tan\left(\frac{u-v}{2}\right)}\rho(u)\rho'(v)dudv + o(t^0). \tag{187}$$

### A.2.2 Sums arising in quantum quench dynamics

We consider

$$\Omega_2(t) = \frac{8}{L^2}\sum_{i \neq j} \frac{\sin q_i \sin q_j}{(\cos q_i - \cos q_j)^2} e^{2it(\overline{\varepsilon}(q_i) - \overline{\varepsilon}(q_j))}$$

$$\tilde{\Omega}_2(t) = \frac{8}{L^2}\sum_{i \neq j} \frac{\sin q_i \sin q_j}{(\cos q_i - \cos q_j)^2} \frac{f(q_i)}{f(q_j)} e^{2it(\overline{\varepsilon}(q_i) - \overline{\varepsilon}(q_j))}. \tag{188}$$

Again, $\Omega_2(t)$ and $\tilde{\Omega}_2(t)$ diverge as $L$ in the thermodynamic limit. But the coefficient of the divergence does not depend on $t$ and is the same for $\Omega_2(t)$ and $\tilde{\Omega}_2(t)$. The differences $\tilde{\Omega}_2(t) - \Omega_2(0)$ are well-defined in the thermodynamic limit. Using the same approach as in the previous section, one obtains

$$\tilde{\Omega}_2(t) - \Omega_2(0) = -16\pi \int_0^\pi \rho(x)^2 |t\overline{\varepsilon}'(x)|dx - 8\int_0^\pi \int_0^\pi dxdy \, \frac{\sin y}{\cos y - \cos x}\rho'(x)\rho(y)$$

$$+ o(t^0) + \mathcal{O}(L^{-1}). \tag{189}$$

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
