# Peer review of "Finite temperature and quench dynamics in the Transverse Field Ising Model from form factor expansions"

_SciPost Physics, doi:SciPost Phys. 9, 033 (2020)_

## Round 1 · Referee Report · Dirk Schuricht (Referee 1) · 2020-3-31

Strengths

  1. Systematic analysis of the asymptotic behaviour of correlation functions in the transverse-field Ising model.
  2. Detailed derivation of result contained.
  3. Possible generalisations explicitly mentioned.

Weaknesses

  1. Presentation is quite technical.
  2. Discussion of discrepancy in prefactor in (128) could be extended.
  3. Few typographical errors.

Report

The authors analyse form-factor expansions for the order parameter in the transverse-field Ising model. Specifically they consider the two-point function at finite temperature and the one-point function after a quantum quench. By writing the form factors as partial fractions the authors are able to derive systematic expansions for the asymptotic behaviour of these observables. In particular, the results clarify and extend previous works. Furthermore, the authors point out several generalisations to be discussed in the future.

The article is rather technical, which cannot be avoided given the addressed problem. Overall the line of argument is clear, with all details discussed in a suitable manner. I fully support publication.

Requested changes

Suggested changes: 1. Typesetting of brackets, eg, in (131) could be improved. 2. Maybe extend the discussion of the discrepancy of the prefactor shown in Fig. 10.

---

## Round 1 · Referee Report · Anonymous (Referee 2) · 2020-5-4

Strengths

  1. A thorough and systematic study
  2. Especially, the idea of evaluating correlation functions is very intriguing
  3. Builds on earlier works systematically and indicates the possibility of future applications.

Weaknesses

  1. Very technical presentation
  2. Difficult to read.
  3. Parts of the main text may be moved to appendix

Report

Although studies on the zero-temperature dynamics of the paradigmatic transverse Ising chain abound, a thorough and systematic study of the finite temperature dynamical properties of the same, that too going beyond low density, is lacking. This paper indeed addresses the issue at length.

The authors provide analytical studies of (i) the dynamical order parameter two-point function at finite temperature and (ii) the one-point function after a quantum quench, which are supported by numerical verifications. Both the problems are studied in terms of spectral representations using Hamiltonian eigenstates, i.e. sums over form factors and the late time behavior of the quantities are computed.

For semi-local operators, authors use an expansion in the density of particles of the representative state, which enables the computation of leading behavior of the same at late times to all orders in the density. On the other hand, for the correlation function, it has been shown that simultaneous particle-hole excitations of all particles in the representative state contribute and it has been shown the form factor sum is dominated by “mesoscopic” excitations around each single particle.

The paper indeed merits publication and will be very useful to the community working in this vibrant field. Only problem I find that the paper is indeed very technical and at least I found it very difficult to go through it. However, given the nature of the problem addressed in this work, one cannot probably help this. I thus recommend publication and leave it to the authors to consider whether the readability and the presentation of the manuscript can be improved.

Requested changes

Already suggested in the report. Some other books and review
articles on transverse Ising models may be cited for the sake
of completeness.

---

## Round 1 · Referee Report · Anonymous (Referee 3) · 2020-6-6

Strengths

  1. Analytical results on the dynamics of the operators like \sigma^{x}_{l}, which are non-local in the Jordan-Wigner fermions that diagonalize the Hamiltonian (semi-local in the sense described in the text), as presented here, are rather rare. These operators are interesting because they have more generic dynamical behavior even in integrable systems. This leads to an expansion organized by the degrees of the pole, and hence an expansion in the density of particles in the representative state.

  2. The authors present a nice technique to overcome the well-known difficulties of calculating form factors with several singularities (several poles) by breaking it into partial fractions. This gives a

  3. Even though not exact, the results are rigorous and accurate (the latter being confirmed by the numerics).

  4. The formalism goes through for semi-local operators in interacting integrable systems.

Weaknesses

  1. The work draws heavily on constructs and technical ideas from previous works without the necessary (even brief) introductions of those. This makes it hard to penetrate for a general physicist who might want to get an idea of the new physics presented in the paper but does not keep the exact technical knowhow of this very specific branch.

  2. In a few places (for example, on the right-hand side of Eq. 12) symbols are used without specifying their meanings.

  3. Interestingly but unfortunately, the formalism does not seem to be extendable for calculating local operators even for integrable cases in the presence of interactions.

Report

In this work, the authors used the so-called quench action formalism to determine the late-time behavior of local and semi-local operators following a quantum quench. The most important result is capturing the late time dynamics of semi-local operators via form-factor calculation. These operators are hard to calculate analytically when the model is mapped to free fermions because they are non-local in terms of those degrees of freedom, and also technically hard because of having severally more singularities compared to the form factors of the local operators. The work is rigorous and probably contains elements that can be generalized for calculating certain operators in a more generic setup.

The authors employed a generalized version of the Eigenstate Thermalization Hypothesis, which allows one to calculate the late-time asymptotics of local/semi-local operators on a suitably chosen single eigenstate of the final Hamiltonian, which can be determined by employing thermodynamic Bethe ansatz. They applied this both for the zero temperature case and the "finite-temperature" case.

This left me with a small confusion regarding the interpretations of the finite-T result. It appears to me that the authors are saying that starting the quench from a finite T density matrix is equivalent to averaging over a suitable eigenstate of the final Hamiltonian. This equivalence is not quite clear to me, because an initial thermal density matrix will show an initial distribution for all the conserved quantities which will evolve with time, but will retain some fluctuations at late times. However, an eigenstate of the final Hamiltonian will have well-defined values for all the conserved quantities (modulo the degeneracies). Hence I wonder if there is a proof or argument for the late time distribution of all the local operators for being always the same in these two cases. In this context, please note that there can be various alternative sets of conserved quantities other than the momentum densities, say, those which are ranked according to their locality, the most important one being most local.

I also wonder if the formalism can be extended to calculate (in a perturbative sense) semi-local operators for non-integrable interacting systems.

Requested changes

No specific request.

---

## Round 1 · Referee Report · Anonymous (Referee 4) · 2020-6-30

Strengths

  1. Analytic results on dynamics of the oder parameter at finite temperature and that after a quantum quench are obtained.
  2. The results are asymptotically exact.
  3. The analytic method developed in this work may creates new possibility for computing correlation functions and expectation values of so-called semi-local operators in non-interacting and interacting integrable models.

Weaknesses

  1. The description is so technical that it is difficult for non-experts on this subject to read.
  2. No description on the organization of the paper in Introduction.
  3. There are a few typos.

Report

It has been difficult to compute time-dependent behaviors of the order parameter in the one-dimensional transverse Ising model in spite of its integrability. In fact, to the best of my knowledge, no exact result has been known on the correlation function as well as the expectation value of the order parameter as functions of time in the thermodynamic limit. The difficulty originates in the fact that one has to take into account the matrix elements of an operator between states belonging to different fermion-parity sectors. The authors of this manuscript resolved this difficulty by making use of the typicality ansatz and partial fraction decompositions.

Although the algebra is quite hard to follow, the results presented here should be valuable to those who study the dynamics of integrable and non-integrable quantum many-body systems.

Requested changes

  1. The organization of the paper should be described in the last section of Introduction.
  2. Correct typos as follows. a) Equation (19): \xi_T -> \xi_L b) Below Eq. (39): Fig. 3.1.1 -> Fig. 1 c) Equation (80): dk -> dx d) The line after Eq. (136): computing -> computed e) Equation (140): the factor C must be inserted in front of exp. f) The second line of Sec. 5.1.2: ‘two’ should be removed. g) The first line after the caption of Fig. 11: T -> \rho_{\beta}

---

## Round 2 · Referee Report · Anonymous (Referee 3) · 2020-7-13

Report

The authors have tried and partially succeeded in answering some of the questions I had. I stick to my previous recommendation of publishing the article in Sci Post.

---

## Round 2 · Referee Report · Anonymous (Referee 2) · 2020-8-5

Report

This paper is certainly publishable in the journal.

Requested changes

Authors have endeavoured to improve the quality of the presentation of the work. I recommend its publication in
the present from...

---

## Round 2 · Referee Report · Anonymous (Referee 4) · 2020-8-5

Report

In this revised version, the introduction is modified considerably so that it includes the outline of the paper and the summary of the main results. Due to this modification, the manuscript is improving in presentation. I now recommend this manuscript for publication.

Requested changes

  1. With regard to eq. (14), give the definition of h and \varepsilon(x) or mention that they are defined in Section 2.
  2. Correct next typos:
  3. 2nd line in the 2nd paragraph of Section 5.1: ‘Section 5.2’ -> ‘Section 3.2’
  4. Caption of Figure 16: h = 0.5 -> h = 1.5

---

## Round 2 · Author Response

We are grateful to the three referees for their careful reading of the manuscript and helpful comments. All three referees note the significant progress made in our work in calculating (dynamical) correlation functions of semi-local operators at finite temperature and after quantum quenches and recommend publication.

In the following we respond to the various points raised.

---Reply to the second referee---

We thank the referee for their report and constructive comments. The referee's main concern is that the presentation of our manuscript is rather technical. On the other hand, as they go on to point out, given the nature of the subject matter this is difficult to avoid. In light of the referee's comments we have made additional efforts to improve the readability of our manuscript. In particular we have added a summary of our main results at the end of the introduction, which hopefully will make it easier for readers to navigate their way through the technical parts that follow. We have also improved the text in the Appendix.

---Reply to the third referee---

We thank the referee for their report and constructive comments. The referee did not request specific changes, but raised several questions to which we now reply in turn.

-The referee notes that ''Interestingly but unfortunately, the formalism does not seem to be extendable for calculating local operators even for integrable cases in the presence of interactions.'' and considers this a weakness of our work. Here we beg to differ: this is not a weakness, but a simply consequence of fundamental differences in the properties of Lehmann representations of local and semi-local operators. Which states dominate the spectral sums of response functions is ultimately a physical property, and these are simply different between local and semi-local operators. As is already mentioned in our manuscript, local operators will be the subject of a separate publication.

-The referee expresses ''a small confusion regarding the interpretations of the finite-T result.'' We discuss two completely separate physical problems: (i) Finite-temperature dynamical (time-dependent) spin-spin correlation functions and (ii) Time evolution of the order parameter after a quantum quench of the transverse field (from $h_0$ to $h$), where the initial state is taken to be the ground state at field $h_0$. These two unrelated problems can both be formulated in terms of spectral representations, and the calculations required to extract the dynamics turn out to be technically similar. The spectral representation in the quench case, and the resulting averaging in the steady state are based on the quench-action approach co-developed by one of us. There is no problem with applying it to problems where the initial density matrix is thermal, but we do not do this here.

  • The referee wonders if the formalism can be extended to calculate (in a perturbative sense) semi-local operators for non-integrable interacting systems. We agree that this is an interesting question that should be addressed in future work. The current state of the art is the methodology developed in our manuscript, and in our view the most immediate application/generalization is to interacting integrable theories such as the Lieb-Liniger model.

---Reply to the fourth referee---

We thank the referee for their report, pointing out a number of typos, and constructive comments. We have corrected the typos mentioned by the referee (as well as some others). We have followed the referee's suggestion and added an extended outline of the paper and summary of our key results at the end of the introduction. We hope this will help in making our work more accessible.

The referee comments that ''The description is so technical that it is difficult for non-experts on this subject to read.'' We are well aware of this issue. However, as the content of our paper is inherently technical we are afraid there is a limit to how accessible we can make our work to a wider audience.

---

## Round 3 · Author Response

We thank the referees again for their careful reading of the manuscript, and for recommending publication.
We corrected the few typos noticed by one of the three referees, which were the only requested changes.

---

## Editorial Decision

published